# An Image is Worth Multiple Words: Learning Object Level Concepts using Multi-Concept Prompt Learning

## Abstract

Textural Inversion, a prompt learning method, learns a singular embedding for a new "word" to represent image style and appearance, allowing it to be integrated into natural language sentences to generate novel synthesised images. However, identifying and integrating multiple object-level concepts within one scene poses significant challenges even when embeddings for individual concepts are attainable. This is further confirmed by our empirical tests. To address this challenge, we introduce a framework for *Multi-Concept Prompt Learning (MCPL)*, where multiple new "words" are simultaneously learned from a single sentence-image pair. To enhance the accuracy of word-concept correlation, we propose three regularisation techniques: *Attention Masking (AttnMask)* to concentrate learning on relevant areas; *Prompts Contrastive Loss (PromptCL)* to separate the embeddings of different concepts; and *Bind adjective (Bind adj.)* to associate new "words" with known words. We evaluate via image generation, editing, and attention visualisation with diverse images. Extensive quantitative comparisons demonstrate that our method can learn more semantically disentangled concepts with enhanced word-concept correlation. Additionally, we introduce a novel dataset and evaluation protocol tailored for this new task of learning object-level concepts.

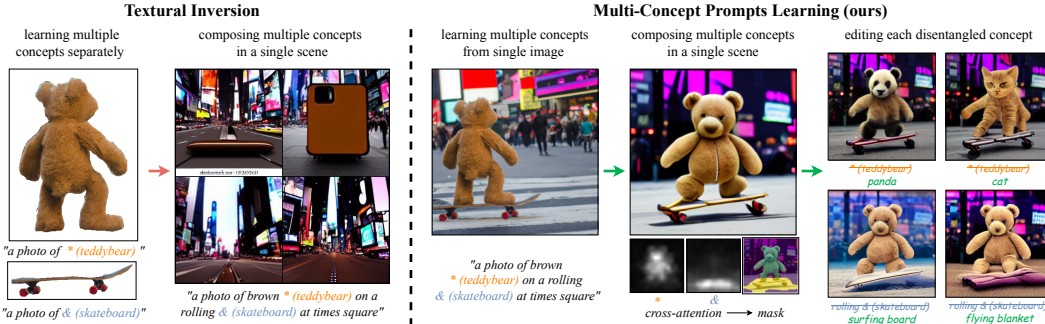

Figure 1: **Multi-concepts learning and composition with previous vs. our approach.** Textural Inversion (left) can only learn a single concept from each image and fails at composing multiple ones. In contrast, our method (right) can learn, compose, and edit multiple concepts simultaneously. The learning input consists of image(s) accompanied by descriptive sentences with learnable prompts, represented as coloured pseudo words. The average cross-attentions and the corresponding mask of the learned prompts denote a disentangled and precise prompt-concept correlation.

## 1 Introduction

In nurseries, toddlers are shown pictures to learn new things. Teachers talk about each picture using sentences with new ideas, like sentences with unfamiliar words. In the Figure 1 right (ours) example, the describing sentence for the images is: *"a photo of brown * on a rolling & at time square"*. Here, *"* (teddy bear)"* and *"& (skateboard)"* are the unfamiliar concepts to be learned.

This way of **learning with images and simple natural language descriptions** is more economical and preferred over the current method of teaching machines using detailed contours and masks.

Recent research (Gal et al. (2022); Ruiz et al. (2022)) shows that the appearance and style of an image can be encapsulated as a cohesive concept via a learned prompt ("word"). The textural embedding of this new prompt is optimised in the frozen embedding space of a pre-trained text-to-image diffusion model to reconstruct several example input images. The concept conveyed by the learned prompt can then be composed into natural language sentences to generate or edit various novel scenes. Despite the significant interest in object-level image editing, (Wu et al., 2020; Meng et al., 2021; Hertz et al., 2022), Gal et al. (2022) points out that recent prompt learning methods struggle with *learning and composing multiple prompts within multi-object scenes* (Figure 1 left).

Break-A-Scene (Avrahami et al., 2023), a mask-based method, recently achieved SoTA in multi-concept prompt learning. Yet, learning object-level concepts using only natural language descriptions, without precise object segmentation, remains largely unexplored. In this work, we start with a motivational study that confirm while applying careful samplings such as manual masking or cropping yields distinct embeddings, object-level learning and editing **with only text-guidance** remains challenging. Motivated by this finding, we introduce *Multi-Concept Prompt Learning (MCPL)* Figure 2 (Top) for **mask-free text-guided learning of multiple prompts from one scene**.

However, without further assumptions on the embedding relationships, jointly learning multiple prompts is problematic. The model may disregard the semantic associations and instead prioritise optimising multiple embedding vectors for optimal image-level reconstruction. To enhance the accuracy of prompt-object level correlation, we propose the following regularisation techniques: 1) To ensure a concentrated correlation between each prompt-concept pair, we propose *Attention Masking (AttnMask)*, restricting prompt learning to relevant regions defined by a cross-attention-guided mask. 2) Recognising that multiple objects within a scene are semantically distinct, we introduce *Prompts Contrastive Loss (PromptCL)* to facilitate the disentanglement of prompt embeddings associated with multiple concepts. 3) To further enable accurate control of each learned embedding, we bind each learnable prompt with a related descriptive adjective word, referred to as *Bind adj.*, that we empirically observe has a strong regional correlation. The middle and bottom row of Figure 2 illustrates the proposed regularisation techniques.

In this work we implement our proposed method based on Textural Inversion by Gal et al. (2022), but the method can be adapted to other prompt learning methods such as Dreambooth by Ruiz et al. (2022). To our knowledge, our technique is the first to address the novel and challenging problem of learning and composing multiple concepts within multi-object scenes. To evaluate this task, we assembled datasets of multi-concept images featuring a total of 16 categories of object-level concepts. These datasets include both natural images, familiar to the pre-trained model, and out-of-distribution biomedical images, each equipped with object-level masks. We evaluate and demonstrate that our framework enables enhanced precision in object-level concept learning, synthesis, editing, quantification, and understanding of relationships between multiple objects, as exemplified in Figure 1 (right) and further illustrated in Figure 9. Through extensive quantitative analysis of approximately 4000 learned object-level embeddings, using both t-SNE and four robust, pre-trained text/image embedding spaces, we validate that our method excels in discerning semantically distinct object-level concepts, ensuring enhanced prompt-to-concept correlation.

## 2  RELATED WORKS

**Prompt learning for image concept inversion.** Prompt tuning, first proposed by Lester et al. (2021), has been utilised to expedite the tuning of large language models for downstream tasks. Jia et al. (2022); Zhou et al. (2022) further extended this approach to vision-language models such as CLIP (Radford et al. (2021)). In the context of text-guided image synthesising, prompt learning would enable connecting the appearance and style of an unseen image to a learnable prompt and transferring to newly generated images, as demonstrated by Textural Inversion Gal et al. (2022) and DreamBooth Ruiz et al. (2022). Addressing multi-concepts, Kumari et al. (2023) fine-tune cross-attention layers using single-concept images for better composition, while Break-A-Scene (Avrahami et al., 2023) employs ground-truth object segmentation for improved learning, yet both approaches depend on carefully selected images or masks.

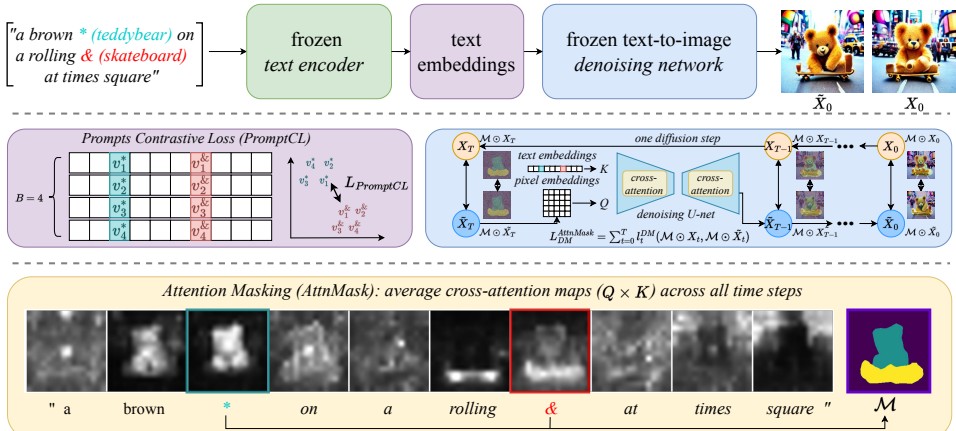

Figure 2: **Method overview.** *MCPL* takes a sentence (top-left) and a sample image (top-right) as input, feeding them into a pre-trained text-guided diffusion model comprising a text encoder $c_\phi$ and a denoising network $\epsilon_\theta$. The string's multiple prompts are encoded into a sequence of embeddings which guide the network to generate images $\tilde{X}_0$ close to the target one $X_0$. MCPL focuses on learning multiple learnable prompts (coloured texts), updating only the embeddings $\{v^*\}$ and $\{v^\&\}$ of the learnable prompts while keeping $c_\phi$ and $\epsilon_\theta$ frozen. We introduce *Prompts Contrastive Loss (PromptCL)* to help separate multiple concepts within learnable embeddings. We also apply *Attention Masking (AttnMask)*, using masks based on the average cross-attention of prompts, to refine prompt learning on images. Optionally we associate each learnable prompt with an adjective (e.g., "brown" and "rolling") to improve control over each learned concept, referred to as *Bind adj.*

**Mask and text-driven local image editing.** In the context of diffusion mode, Meng et al. (2021) first proposed SDEdit for mask-guided image-to-image style translation. Lugmayr et al. (2022) developed RePaint to enable mask-guided local image editing. Avrahami et al. (2022) further conditioned local editing with text condition. These methods use manual masks prior to guide local image editing. A set of recent works showed that text-guided local object-level editing can be achieved without using a mask prior but instead the attention-derived masks (Hertz et al. (2022); Tumanyan et al. (2023); Patashnik et al. (2023)). The success of these approaches heavily relies on the accurate text-concept semantic correlation in the pre-trained model and is limited to in-distribution concepts.

**Disentangled per-concept image editing.** Interpretable and disentangled per-concept image manipulation has garnered significant interest in the literature on Generative Adversarial Networks (GANs). Traditional approaches often focus on layer-wise or channel-wise control within a pre-trained generator network. The goal is to identify and modify a subset of parameters responsible for specific concepts (Brock et al., 2018; Karras et al., 2020; Wu et al., 2020). Although our work is not centred on GAN-based approaches, we emphasise that we directly optimise multiple embeddings rather than network parameters. This methodology has been shown to better adapt to unseen and novel concepts by Gal et al. (2022).

## 3 METHODS

In this section, we outline the preliminaries in Section 3.1 and present a motivational study in Section 3.2. These tests show the challenges of applying current methods in text-guided learning of multiple prompts from one scene. Inspired by these results, we introduce the *Multi-Concept Prompt Learning (MCPL)*. To address the multi-object optimisation challenge tandem with a single image-level reconstruction goal, we propose several regularisation techniques in Section 3.4.

### 3.1 PRELIMINARIES: PROMPT LEARNING IN TEXT-TO-IMAGE DIFFUSION MODEL

**Text-guided diffusion models** are probabilistic generative models trained to approximate the training data distribution through a process of incremental denoising from Gaussian random noise, conditioned on text embeddings. Specifically, a denoising network $\epsilon_\theta$ is trained to map an initial noise map $\epsilon \sim \mathcal{N}(\mathbf{0}, \mathbf{I})$ and conditional textual embedding $v = c_\phi(p)$ to generate images $\tilde{x}$ close to the

target one $x$. Here $c_\phi$ is the text encoder and $p$ is the text prompt. To enable sequential denoising, $c_\phi$ and $\epsilon_\theta$ are jointly optimised to minimize the loss:

$$L_{DM} = L_{DM}(x, \tilde{x}) := E_{x_0, \epsilon \sim N(0,I), t \sim \text{Uniform}(1,T)} \| \epsilon - \epsilon_\theta(x_t, t, c_\phi(p)) \|^2, \qquad (1)$$

where $x_t$ is obtained by adding noise to the initial image $x_0$ at a given time step $t$ in the set $T$. During inference, the pre-trained model iteratively eliminates noise from a new random noise map to generate a fresh image. Our work builds on Latent Diffusion Models (LDMs) (Rombach et al., 2022), which operate on the latent representation $z = \mathcal{E}(x)$ provided by an encoder $\mathcal{E}$.

The prompt learning method by (Gal et al. (2022)) is aimed at identifying the text embedding $v^*$ for a new prompt $p^*$ in a pre-trained text-guided diffusion model. Given a few (3-5) example images representing a specific subject or concept, the method optimises $v^*$ in the frozen latent space of a pre-trained text encoder $c_\phi$. The objective is to generate an image via the denoising network $\epsilon_\theta$ that closely resembles the example images when conditioned on $v^*$. The optimisation is guided by the diffusion model loss defined in equation 1, updating only $v^*$ while keeping $c_\phi$ and $\epsilon_\theta$ frozen. Our training approach aligns with the Textural Inversion strategy outlined in Appendix A.11.

**Cross-attention layers** play a pivotal role in directing the text-guided diffusion process. Within the denoising network, $\epsilon_\theta$, the textual embedding, $v = c_\phi(p)$, interacts with the image embedding, $z = \mathcal{E}(x)$, via the cross-attention layer. Here, $Q = f_Q(z)$, $K = f_K(v)$, and $V = f_V(v)$ are acquired using learned linear layers $f_Q, f_K, f_V$. As Hertz et al. (2022) highlighted, the per-prompt cross-attention maps, $M = \text{Softmax}(QK^T/\sqrt{d})$, correlate to the similarity between $Q$ and $K$. Therefore the average of the cross-attention maps over all time steps reflects the crucial regions corresponding to each prompt word, as depicted in Figure 2. In this study, the per-prompt attention map is a key metric for evaluating the correlation between prompt and concept. Our results will show that without adequate constraints, the attention maps for newly learned prompts often lack consistent disentanglement and precise prompt-concept correlation.

### 3.2 MOTIVATIONAL STUDY: IS IMAGE-LEVEL PROMPT LEARNING SUFFICIENT FOR OBJECT-LEVEL MULTI-CONCEPT LEARNING?

**Do multiple distinct embeddings arise from the same image?** To understand the challenges in learning and composing multiple concepts, we explored whether *Textural Inversion* can discern semantically distinct concepts from processed images, each highlighting a single concept. Following Wu et al. (2020), we used images with manual masks to isolate concepts, as seen in Figure 3 (left). We applied *Textural Inversion* to these images to learn embeddings for the unmasked or masked images. Our findings indicate that when focusing on isolated concepts, *Textural Inversion* can successfully learn distinct embeddings, as validated by the generated representations of each concept.

**Is separate learning of concepts sufficient for multi-object image generation?** While separate learning with carefully sampled or masked images in a multi-object scene deviates from our objective, it is valuable to evaluate its effectiveness. Specifically, we use Textural Inversion to separately learn concepts like "ball" and "box" from carefully cropped images, as shown in Figure 3 (second column). We then attempt to compose images using strings that combine these concepts, such as "a photo of a green ball on orange box." Our results indicate that the accurate composition of multi-object images remains challenging, even when individual concepts are well-learned.

### 3.3 MULTI-CONCEPT PROMPT LEARNING (MCPL)

Our motivational study confirm that: 1) multiple unique embeddings can be derived from a single multi-concept image, albeit with human intervention, and 2) despite having well-learned individual concepts, synthesizing them into a unified multi-concept scene remains challenging. To address these issues, we introduce the Multi-Concept Prompt Learning (MCPL) framework. MCPL modifies Textural Inversion to enable simultaneous learning of multiple prompts within the same string. In specific, MCPL learn a list of multiple embeddings $\mathcal{V} = [v^*, \ldots, v^\&]$ corresponds to multiple new prompts $\mathcal{P} = [p^*, \ldots, p^\&]$. The optimisation is still guided by the image-level $L_{DM}$, but now updating $\{v^*, \ldots, v^\&\}$ while keeping $c_\phi$ and $\epsilon_\theta$ frozen. The MCPL algorithm is outlined in Appendix A.11, Algorithm 2. Recognising the complexity of learning multiple embeddings with a single image-generation goal, we propose three training strategies: 1) *MCPL-all*, a naive approach

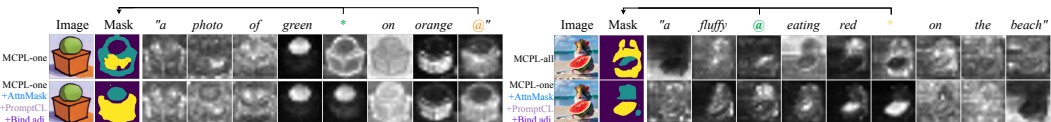

Figure 3: *minor-update* **Motivational Study with "Watch Face-Band" and "Ball-Box" Images.**
Left: Embeddings are learned using *Textural Inversion (T.I.)* on both multi-concept (unmasked) and single-concept (masked) images. Right: Concepts of "ball" and "box" are learned and composed using different methods: *T.I.*, which crops and learns each concept separately; *MCPL-one*, learning both concepts jointly from uncropped examples with a single string; and *MCPL-diverse* accounting for per-image specific relationships. Refer to Appendix Section A.9 for more results.

that learns embeddings for all prompts in the string (including adjectives, prepositions and nouns. etc.); 2) *MCPL-one*, which simplifies the objective by learning single prompt (nouns) per concept; 3) *MCPL-diverse*, where different strings are learned per image to observe variances among examples. Preliminary evaluations of *MCPL-one* and *MCPL-diverse* methods on the "ball" and "box" multi-concept task are shown in Figure 3. Our findings indicate that *MCPL-one* enhance the joint learning of multiple concepts within the same scene over separate learning. Meanwhile, *MCPL-diverse* goes further by facilitating the learning of intricate relationships between multiple concepts.

**Limitations of plain MCPL.** Our primary aim is to facilitate accurate interpretation and modification of multi-concept scenes. To evaluate object-level prompt-concept correlation, we visualise the average cross-attention maps for each prompt. As depicted in Figure 4, both *MCPL-one* and *MCPL-all* inadequately capture this correlation, especially for the target concept. These results suggest that *naively extending image-level prompt learning techniques (Gal et al., 2022) to object-level multi-concept learning poses optimisation challenges*, notwithstanding the problem reformulation efforts discussed in Section 3.3. Specifically, optimising multiple object-level prompts based on a single image-level objective proves to be non-trivial. Given the image generation loss equation 1, prompt embeddings may converge to trivial solutions that prioritize image-level reconstruction at the expense of semantic prompt-object correlations, thereby contradicting our objectives. In the next section, we introduce multiple regularisation terms to overcome this challenge.

Figure 4: *(reposition)* **Enhancing object-level prompt-concept correlation in MCPL using the proposed *AttnMask*, *PromptCL* and *Bind adj.* regularisation techniques.** We compare our best results of *MCPL-one* applying all regularisation terms against the plain *MCPL-one*, using a "Ball and Box" example (left) and the plain *MCPL-all*, using a "Hamster and Watermelon" example (right). We use the average cross-attention maps and the *AttnMask* to assess the accuracy of correlation.

### 3.4 REGULARISING THE MULTIPLE OBJECT-LEVEL PROMPTS LEARNING

**Encouraging focused prompt-concept correlation with Attention Masking (*AttnMask*).** Previous results show plain *MCPL* may learn prompts focused on irrelevant areas. To correct this, we apply masks to both generated and target images over all the denoising steps (Figure 2, middle-right). These masks, derived from the average cross-attention of learnable prompts (Figure 2, bottom-row), constrain the image generation loss (equation 1) to focus on pertinent areas, thereby improving prompt-concept correlation. To calculate the mask, we compute for each learnable prompt $p \in \mathcal{P}$ the average attention map over all time steps $\overline{M}^p = 1/T \sum_{t=1}^{T} M_t^p$. We then apply a threshold to produce binary maps for each learnable prompt, where $B(\overline{M}^p) := \{1 \text{ if } M^p > k, 0 \text{ otherwise}\}$ and $k = 0.5$ throughout all our experiments. For multiple prompt learning objectives, the final mask $\mathcal{M}$ is a union of multiple binary masks of all learnable prompts $\mathcal{M} = \bigcup_{p \in \mathcal{P}} B(M^p)$. We compute

the Hadamard product of $\mathcal{M}$ with $x$ and $\tilde{x}$ to derive our masked loss $L_{DM}^{AttnMask}$ as equation 2. Our *AttnMask* is inspired by Hertz et al. (2022), but a reverse of the same idea, where the *AttnMask* is applied over the pixel-level loss equation 1 to constrain the prompt learning to only related regions.

$$L_{DM}^{AttnMask} = L_{DM}(\mathcal{M} \odot x, \mathcal{M} \odot \tilde{x}), \tag{2}$$

**Encouraging semantically disentangled multi-concepts with Prompts Contrastive Loss (*PromptCL*).** *AttnMask* focuses the learning of multiple prompts on the joint area of target objects, eliminating the influence of irrelevant regions like the background. However, it doesn't inherently promote separation between the embeddings of different target concepts. Leveraging the mutual exclusivity of multiple objects in a scene, we introduce a contrastive loss in the latent space where embeddings are optimised. Specifically, we employ an InfoNCE loss Oord et al. (2018), a standard in contrastive and representation learning, to encourage disentanglement between groups of embeddings corresponding to distinct learnable concepts (Figure 2, middle-left).

Concretely, at each learning step as described in Algorithm 2, a mini-batch $B$ minor augmented (e.g. with random flip) example images are sampled, with $N$ learnable prompts/concepts for each image, yields a set of $BN$ embeddings, $\{v_b^n\}_{b=1, n=1}^{B, N}$. Then, the similarity between every pair $v_i$ and $v_j$ of the $BN$ samples is computed using cosine similarity, i.e. $sim(v_i, v_j) = v_i^T . v_j / ||v_i|| ||v_j||$. Given our goal is to differentiate the embeddings corresponding to each prompt, we consider the embeddings of the same concept as positive samples while the others as negative. Next, the contrastive loss $l_{i,j \in B}^{\eta}$ for a positive pair $v_i^{\eta}$ and $v_j^{\eta}$ of each concept $\eta \in N$ (two augmented views of the example image) is shown in the equation 3, where $\tau$ is a temperature parameter following Chen et al. (2020). The contrastive loss is computed for $BN$ views of each of the $N$ learnable concepts. The total contrastive loss $L_{PromptCL}$ is shown in equation 4 (left).

$$l_{i,j \in B}^{\eta} = -log\left(\frac{exp(sim(v_i^{\eta}, v_j^{\eta}))/\tau}{\sum_{\eta=1}^{N} \sum_{j=1, j \neq i}^{B} exp(sim(v_i^{\eta}, v_j^{\eta})/\tau)}\right) \tag{3}$$

$$L_{PromptCL} = \frac{1}{N} \frac{1}{B} \sum_{\eta=1}^{N} \sum_{i=1}^{B} l_{i,j \in B}^{\eta}, \qquad L_{PromptCL}^{adj} = \frac{1}{NM} \frac{1}{B} \sum_{\eta=1}^{NM} \sum_{i=1}^{B} l_{i,j \in B}^{\eta} \tag{4}$$

**Enhance prompt-concept correlation by binding learnable prompt with the adjective word (*Bind adj.*).** An additional observation from the misaligned results in Figure 38 reveals that adjective words often correlate strongly with specific regions. This suggests that the pre-trained model is already adept at recognising descriptive concepts like colour or the term "fluffy." To leverage this innate understanding, we propose to optionally associate one adjective word for each learnable prompt as one positive group during the contrastive loss calculation. In particular, consider $M$ adjective words associated with $N$ learnable prompts. Then the positive pair $v_i^{\eta}$ and $v_j^{\eta}$ of each concept is sampled from $\eta \in MN$ instead of $N$. Therefore the contrastive loss is now computed for $BNM$ views of each of the $N$ learnable concepts. The resulting total contrastive loss $L_{PromptCL}^{adj}$ is detailed in equation 4 (right). We scale $L_{PromptCL}^{adj}$ with a scaling term $\gamma$ and add with $L_{DM}^{AttnMask}$ (equation 2), for them to have comparable magnitudes, resulting our final loss in equation 5.

$$L = L_{DM}^{AttnMask} + \gamma L_{PromptCL}^{adj}, \tag{5}$$

**Implementation details.** Unless otherwise noted, we retain the original hyper-parameter choices of LDM (Rombach et al., 2022). All learnable embeddings were initialised by the encoding of each pseudo word, such as "*". Our experiments were conducted using a single V100 GPU with a batch size of 4. The base learning rate was set to 0.005. Following LDM, we further scale the base learning rate by the number of GPUs and the batch size, for an effective rate of 0.02. On calculating $L_{PromptCL}$, we apply the temperature and scaling term $(\tau, \gamma)$ of $(0.2, 0.0005)$ when *AttnMask* is not applied, and $(0.3, 0.00075)$ when *AttnMask* is applied. All results were produced using 6100 optimisation steps. We find that these parameters work well for most cases.

# 4 RESULTS

## 4.1 ASSESSING REGULARISATION TERMS WITH CROSS-ATTENTION

We start with assessing our proposed regularisation terms on improving the accuracy of semantic correlations between prompts and concepts. We visualise the cross-attention and segmentation masks, as shown in Figure 4. Our visual results suggest that incorporating all of the proposed regularisation terms enhances concept disentanglement, whereas applying them in isolation yields suboptimal outcomes (refer to full ablation results in Appendix A.10). Moreover, the results demonstrate that *MCPL-one* is a more effective learning strategy than *MCPL-all*, highlighting the importance of excluding irrelevant prompts to maintain a focused learning objective.

## 4.2 QUANTITATIVE EVALUATIONS

We collect both in-distribution natural images and out-of-distribution biomedical images over 16 object-level concepts, with all images containing multiple concepts and object-level masks (see examples in Figure 5). We collected 40 images for each concept, full details of dataset preparation in Appendix A.12. We compare three baseline methods: 1) *Textural Inversion* applied to each masked object serving as our best estimate for the unknown disentangled "ground truth" embedding. 2) *Break-A-Scene (BAS)*, the state-of-the-art (SoTA) mask-based multi-concept learning method, serves as a performance upper bound, though it's not directly comparable. 3) *MCPL-all* as our naive adaptation of the *Textural Inversion* method to achieve the multi-concepts learning goal. For our method MCPL-all and MCPL-one, we examine four variations to scrutinise the impact of the regularisation terms discussed in Section 3.4. It's important to note that, all learning with our method is performed on unmasked images. To assess the robustness of each learning method, we randomly sample four images to learn an embedding, leading to 10 (all MCPL-based methods) or 5 (BAS) learned embeddings per concept. The experiments were executed on a single V100 GPU, with each run taking approximately one hour, resulting in a total computational cost of around 2100 GPU hours (or 87 days on a single GPU). We employed various metrics to evaluate the four methods.

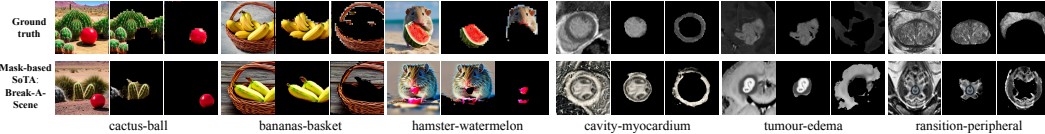

Figure 5: Visualisation of the prepared ground truth examples (top) and the generated images with Break-A-Scene (bottom). Note that BAS requires segmentation masks as input and employs separate segmentation models to produce masked objects, thus serving as a performance upper-bound. See the full 16 concepts dataset in Appendix A.13 and all BAS generated images in Appendix A.6.

**Investigate the disentanglement of learned embeddings with t-SNE.** To assess disentanglement, we begin by visualising the t-SNE projection of the learned features Van der Maaten & Hinton (2008). The results, depicted in Figure 7, encompass both natural and biomedical datasets. They illustrate that our *MCPL-one* combined with all regularisation terms can effectively distinguish all learned concepts compared to all baselines. It's noteworthy that the learned embeddings from both the mask-based 'ground truth' and BAS show less disentanglement compared to ours, attributable to their lack of a specific disentanglement objective, such as the PromptCL loss in MCPL. This finding confirms the necessity of our proposed method.

**Embedding similarity comparing to the estimated "ground truth".** To assess the preservation of per-concept semantic and textural details, we calculate both prompt and image fidelity. This evaluation follows prior research by Gal et al. (2022) and Ruiz et al. (2022), but differently, we perform the calculations at the object level. We compared the masked "ground truth" and generated masked-objects across four embedding spaces. For both BAS and our MCPL variants, we initially learned object-level concepts and then generated masks. Specifically for BAS, we used separate pre-trained segmentation models—MaskFormer (Cheng et al., 2021) for natural images and human-in-the-loop MedSAM (Ma & Wang, 2023) for medical images—to create masked objects (see Figure 5 and Appendix A.1 for details). In contrast, our method employed its own *AttnMask* to generate

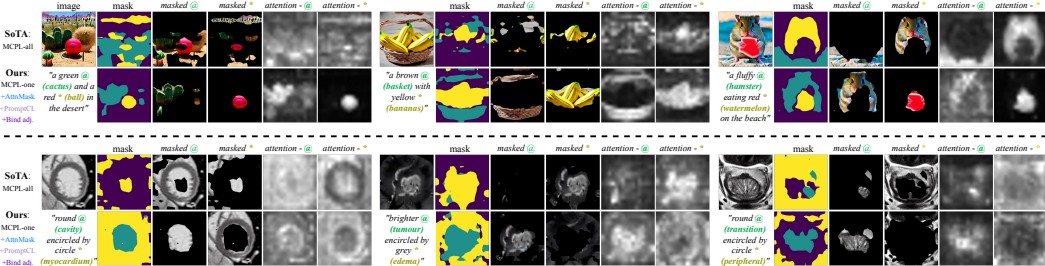

Figure 6: **Visualisation of generated concepts with the "SoTA" and our method. Masks are derived from cross-attentions.** Full ablation results are presented in the Appendix A.8

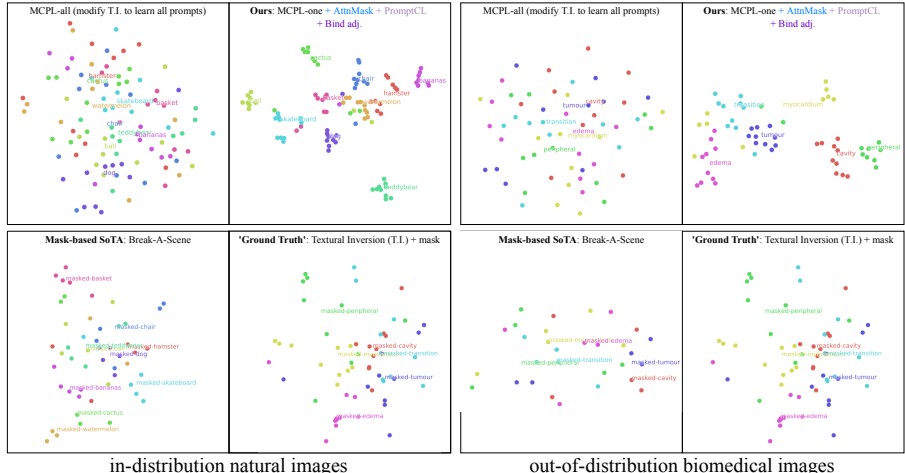

Figure 7: **The t-SNE projection of the learned embeddings**. Our method can effectively distinguish all learned concepts compared to Textural Inversion (MCPL-all), the SoTA mask-based learning method, Break-A-Scene, and the masked 'ground truth' (full results in Appendix A.7).

masked images (as shown in Figure 6). We generated a total of 320 masked objects for each MCPL variant and 160 for the BAS baseline, with 20 (10 for BAS) masked images per concept. Prompt fidelity is determined by measuring the average pairwise cosine similarity between the embeddings learned from the estimated "ground truth" and the generated masked images, in the pre-trained embedding space of BERT (Devlin et al., 2018). For image fidelity, we compare the average pairwise cosine similarity in the pre-trained embedding spaces of CLIP Radford et al. (2021), DINOv1 (Caron et al., 2021) and DINOv2 (Oquab et al., 2023), all based on the ViT-S backbone.

The results in Figure 8 show our method combined with all the proposed regularisation terms can improve both prompt and image fidelity consistently. Our fully regularised version (MCPL-one+CL+Mask) achieved competitive performance compared to the SoTA mask-based method (BAS) on the natural dataset. In the OOD medical dataset, BAS outperformed our method significantly in the DINOv1 embedding space, although the performance was comparable in other spaces. This discrepancy is due to the less accurate object masks in our method compared to BAS, which employs human-in-the-loop MedSAM (Ma & Wang, 2023) for segmentation, as evident in Figure 6 and Figure 5.

### 4.3 APPLICATIONS: IMAGE EDITING OVER DISENTANGLED CONCEPTS.

Finally we demonstrate our method enables more accurate object-level synthesis, editing and quantification (Figure 9 top-left). The framework also has the flexibility to handle per-image specified string to learn the differences concepts within each image, as shown in the top-right example of Figure 9. Furthermore, our method can also learn unknown concepts from challenging out-of-distribution images (Figure 9 bottom-left and right), opening an avenue of knowledge mining from pairs of textbook figures and captions, which are abundantly available on the internet.

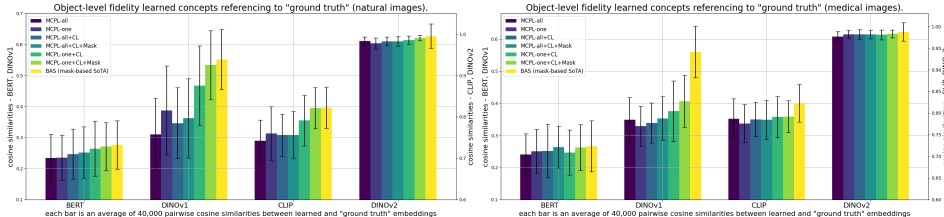

Figure 8: **Embedding similarity in learned object-level concepts compared to masked "ground truth".** We evaluate the embedding similarity of our multi-concept adaptation of Textural Inversion (MCPL-all) and the state-of-the-art (SoTA) mask-based learning method, Break-A-Scene (BAS) by Avrahami et al. (2023), against our regularised versions. The analysis is conducted in both pre-trained text (BERT) and image encoder spaces (CLIP, DINOv1, and DINOv2), with each bar representing an **average of 40,000** pairwise cosine similarities. A comprehensive object-level comparison is available in the Appendix (Section A.5).

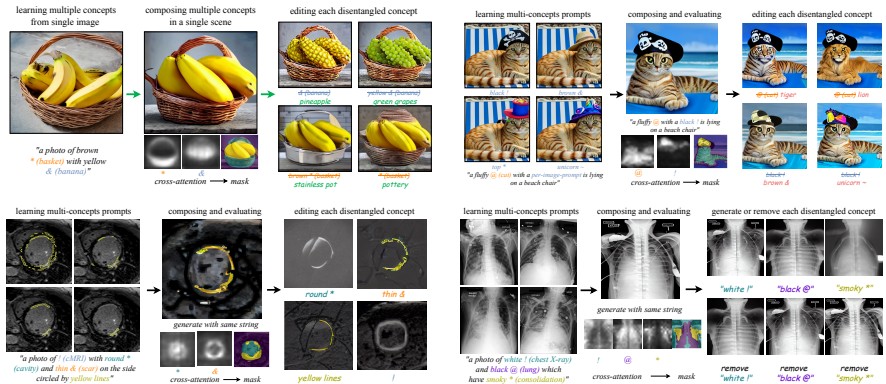

Figure 9: **MCPL learning and composing capabilities.** (top-left) learning and editing multiple concepts with a single string; (top-right) learning per-image different concepts with per-image specified string; (bottom-left) learning to disentangle multiple unseen concepts from cardiac MRI images; (bottom-right) learning to disentangle multiple unseen concepts from chest X-ray images.

## 5    LIMITATIONS AND CONCLUSIONS

We identify the following limitations in our method: (1) Imperfect Masking: Our reliance on image-level text descriptions, instead of segmentation masks, grants flexibility in exploring unknown concepts but results in less precise object boundary optimization. Future research could use our Attn-Mask as an input prompt to segmentation models for mask refinement. (2) Composition Capability: MCPL's composition strength is weaker than BAS, as MCPL doesn't update model parameters, unlike BAS. Integrating MCPL with weight optimization methods like BAS or DreamBooth may enhance performance, albeit at higher computational costs, which is a potential direction for future work. (3) Evaluation Metrics: Current quantification methods in this field (e.g. TI, DB, CD, BAS, and P2P), predominantly rely on prompt/embedding similarity due to the absence of more effective quantification mechanisms without known ground truth. This indicates a need for developing better evaluation metrics in future research. (4) Our method relies on adjectives serving as textual descriptors (e.g., color) to differentiate between multiple concepts. While human-machine interaction using purely linguistic descriptions is generally preferred, challenges arise when two concepts are very similar and lack distinct visual cues in the image. In such cases, our method may struggle, and Break-A-Scene currently offers the best solution.

In conclusion, MCPL is introduced to address the novel challenge of learning multiple concepts using images and simple natural language descriptions. We anticipate that this will pave the way for knowledge discovery through natural language-driven human-machine interaction, leading to advancements in tasks like synthesis, editing, quantification, and a more precise understanding of multi-object relationships at the object level.

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

# A APPENDIX

Here we provide additional results and various ablation studies and implementation details that have not been presented in the main paper.

## CONTENTS

## A.1 BREAK-A-SCENE EXPERIMENTS SETUP

Break-A-Scene (BAS) (Avrahami et al., 2023) learns multiple concepts from images paired with object-level masks. It augments input images with masks to highlight target concepts and updates both textural embeddings and model weights accordingly. BAS introduces 'union sampling', a training strategy that randomly selects subsets of multi-concepts in each iteration to enhance the combination of multiple concepts in generated images, see Figure 11 for an illustration. During inference, BAS employs a pre-trained segmentation model to obtain masked objects, facilitating localized editing.

To fit BAS (Avrahami et al., 2023) into our evaluation protocol, we first learned object-level concepts and then generated masked objects for evaluation, including the following steps:

1. BAS Learning: For each concept pair, we randomly selected 20 images with ground truth segmentations from our dataset for BAS learning, resulting in 20 BAS embeddings per concept.

2. BAS Generation: We then generated 20 images for each concept pair, producing a total of 100 BAS-generated natural images and 60 medical images.

3. Segmentation: For masked object production with BAS, we used different pre-trained segmentation models. MaskFormer (Cheng et al., 2021) was effective for natural images, but segmenting medical images posed challenges due to their out-of-distribution characteristics.

4. Quantitative Evaluation: With the obtained masked objects (20 per concept, see visualizations in Section A.6), we applied the embedding similarity evaluation protocol from Section 4.2 to assess the preservation of semantic and textural details per concept in four embedding spaces.

For segmenting medical images, given the diversity of classes in our dataset, we utilized MedSAM (Ma & Wang, 2023), a state-of-the-art foundation model adapted from SAM (Kirillov et al., 2023)

for the medical domain. MedSAM requires a bounding box for input, making it a multi-step, human-in-the-loop process. We initially assessed segmentation quality from several (up to five) bounding box proposals, as exemplified in Figure 10. MedSAM, despite having a bounding box, cannot fully automate segmentation for all classes. Thus, we employed an additional post-processing step to discern the segmentation of both classes by calculating the difference between the two segmentations. This procedure was applied to all 60 BAS-generated medical images.

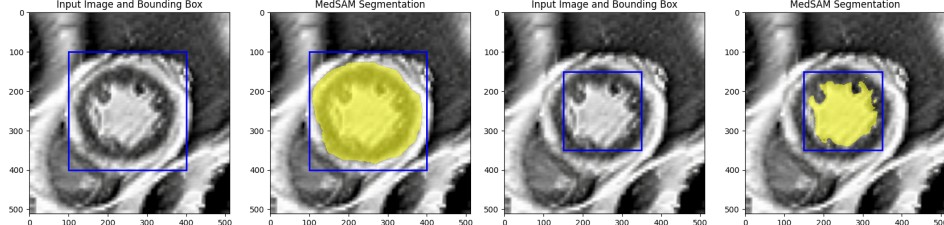

Figure 10: This demonstration shows how MedSAM is used to segment medical images generated by BAS. On the left, MedSAM segmentation with a large bounding box prompt can identify the combined area of the cavity-myocardium classes, but it does not distinguish between the two. On the right, using a smaller bounding box prompt, MedSAM successfully segments the central cavity class. We calculate the difference between the two segments to get the segmentation of the missing myocardium class (outer ring-like pattern).

### A.2 VISUAL COMPARISON OF MCPL-DIVERSE/ONE WITH MASK-BASED APPROACHES

In tasks learning more than two concepts from a single image, we compare MCPL with Break-A-Scene (BAS). Unlike BAS, MCPL neither uses segmentation masks as input nor updates model parameters. To level the playing field, we adopted BAS's 'union sampling' training strategy, which randomly selects subsets of multi-concepts in each iteration. However, lacking mask input, we manually prepared a set of cropped images of each individual concept and randomly selected subsets to combine. This approach, termed *'random crop,'* serves as our equivalent training strategy, see Figure 11 for an illustration. Given that each cropped image has a different number of concepts, we utilized our **MCPL-diverse**, designed to learn varying concepts per image. In Figure 11 and Figure 12 we showcase examples of such tasks against a set competitive baselines.

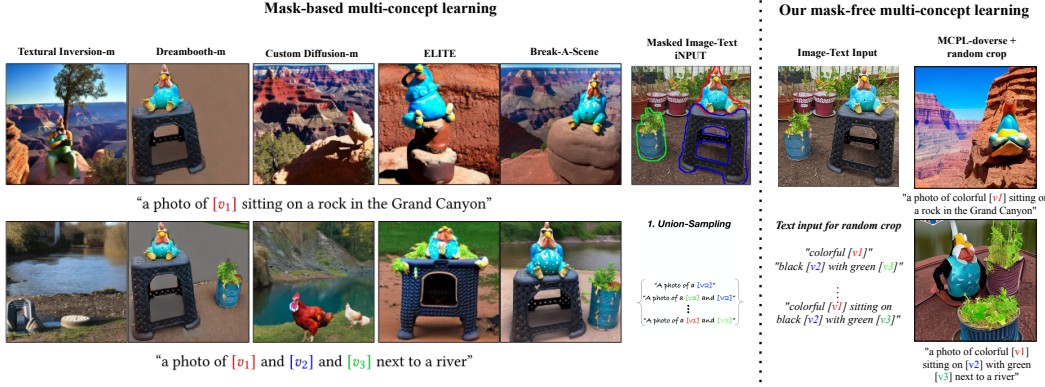

Figure 11: A qualitative comparison between our method (MCPL-diverse) and mask-based approaches: Break-A-Scene (Avrahami et al., 2023), Textural Inversion (Gal et al., 2022) (masked version), DreamBooth (Ruiz et al., 2022) (masked version), Custom Diffusion (Kumari et al., 2023) (masked version) and ELITE (Wei et al., 2023). Our MCPL-diverse, which **neither uses mask inputs nor updates model parameters**, showed decent results, outperforming most mask-based approaches and was closer to SoTA Break-A-Scene. Images modified from Break-A-Scene (Avrahami et al., 2023).

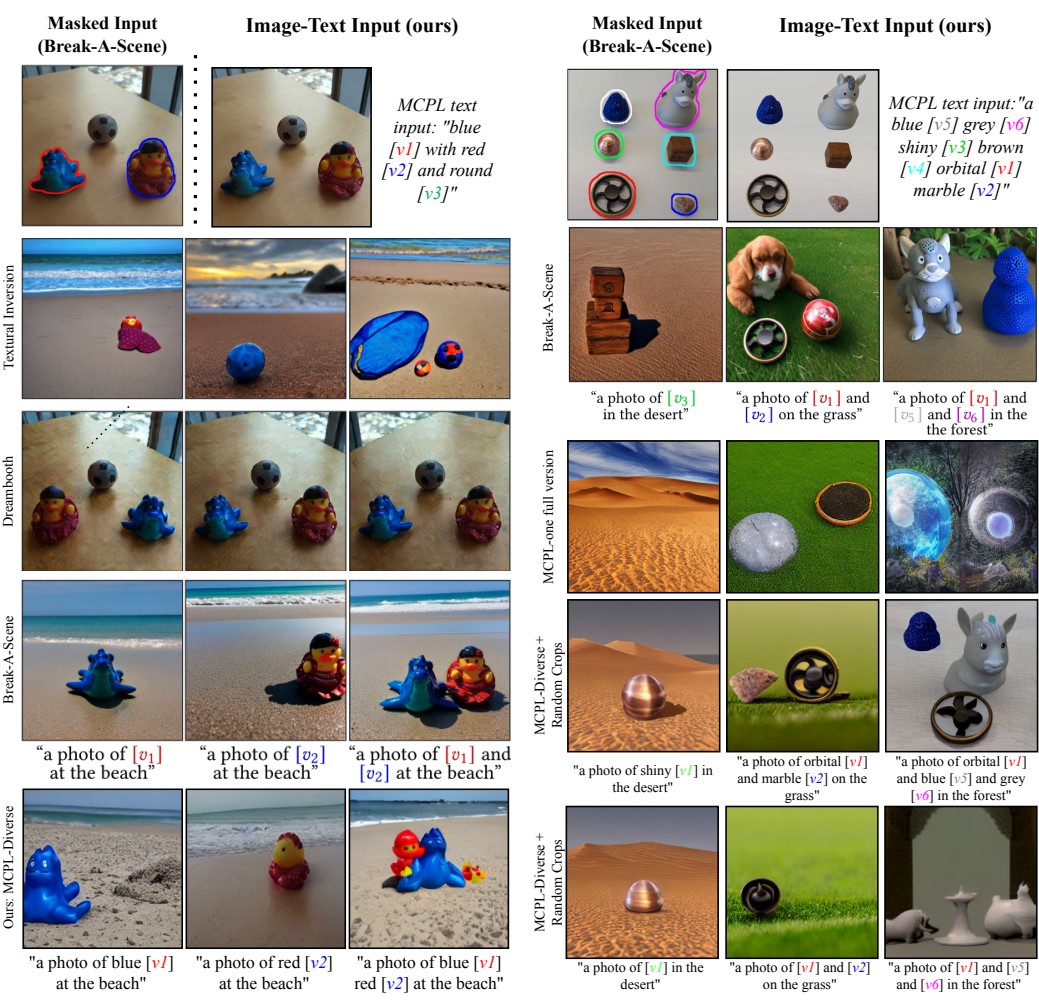

Figure 12: A qualitative comparison between Textural Inversion (Gal et al., 2022), DreamBooth (Ruiz et al., 2022), Break-A-Scene (Avrahami et al., 2023) and our method (MCPL-one and MCPL-diverse). **The left task** learns three concepts from a single image and then composes a subset (two concepts) in a novel scene. Both Textural Inversion and DreamBooth were unsuccessful, while Break-A-Scene, using mask input and model parameter updates, performed best. Our MCPL-diverse, which **neither uses mask inputs nor updates model parameters**, showed decent results, outperforming Textural Inversion and DreamBooth, and was closer to BAS. **The task on the right**, involving learning six concepts from a single image and then composing a subset of $1 \sim 3$ concepts in a new scene, is particularly challenging. Break-A-Scene (Avrahami et al., 2023) has acknowledged limitations in learning more than four concepts. In our study, we evaluated both MCPL-one and MCPL-diverse, the latter employing random crops akin to BAS's 'union sampling' strategy. Our findings reveal that: 1) our equivalent to 'union sampling' effectively enhances results by exposing the model to more concept combinations; 2) similar to BAS, our method also faces challenges with a high number of concepts, but it shows marginally better performance, likely aided by the use of adjectives; 3) as a validation, removing adjectives at inference leads to a noticeable performance drop. Images modified from Break-A-Scene (Avrahami et al., 2023).

A.3  ABLATION STUDY COMPARING MCPL-DIVERSE VERSUS MCPL-ONE IN LEARNING PER-IMAGE DIFFERENT CONCEPT TASKS

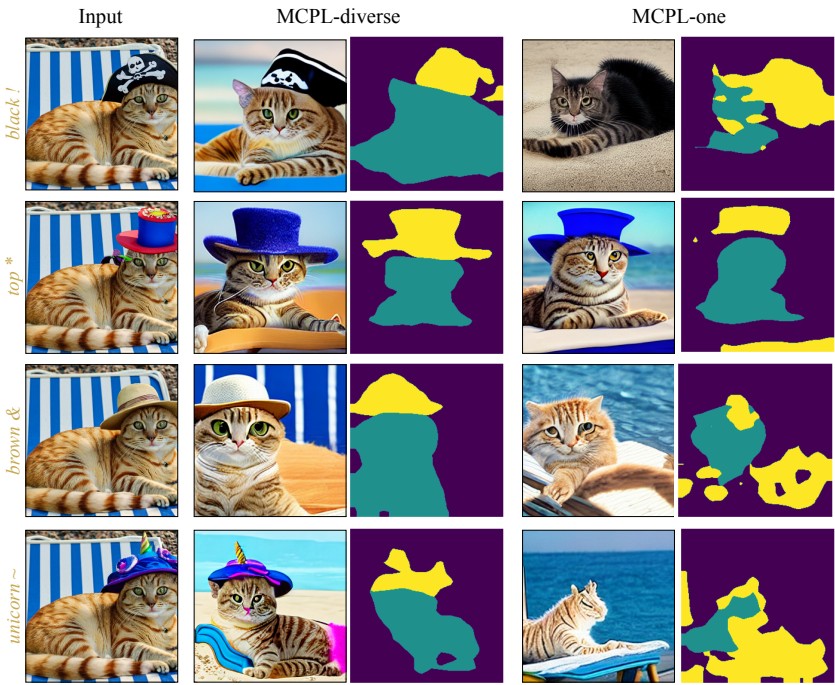

*Learning with per image different prompts*
*"a fluffy @ (cat) with a per-image-prompt is lying on a beach chair"*

Figure 13: Visual comparison of MCPL-diverse versus MCPL-one in learning per-image different concept tasks (cat with different hat example). As MCPL-diverse are specially designed for such tasks, it outperforms MCPL-one, which fails to capture per image different hat styles.

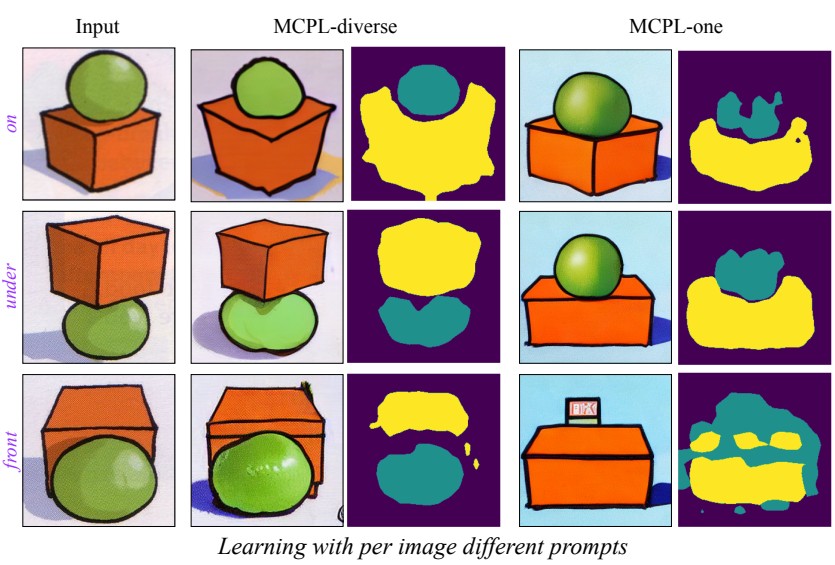

*Learning with per image different prompts*
*"a photo of green * per-image-prompt orange @"*

Figure 14: Visual comparison of MCPL-diverse versus MCPL-one in learning per-image different concept tasks (ball and box relationships example). As MCPL-diverse are specially designed for such tasks, it outperforms MCPL-one, which fails to capture per image different relationships.

### A.4 ABLATION STUDY ON EFFECT OF ADJECTIVE WORDS.

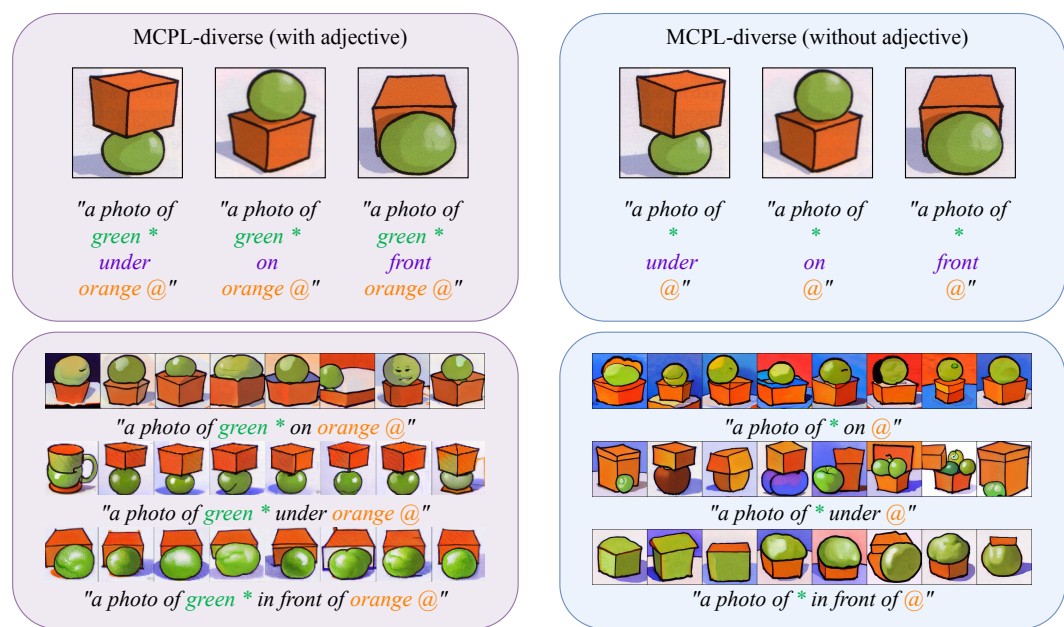

Figure 15: Visual comparison of MCPL-diverse with adjective word versus without adjective word. Adjective words are crucial in linking each prompt to the correct region; without them, the model may struggle for regional guidance and we observe reduced performance.

### A.5 ALL OBJECT-LEVEL EMBEDDING SIMILARITY OF THE LEARNED CONCEPT COMPARED TO THE ESTIMATED "GROUND TRUTH".

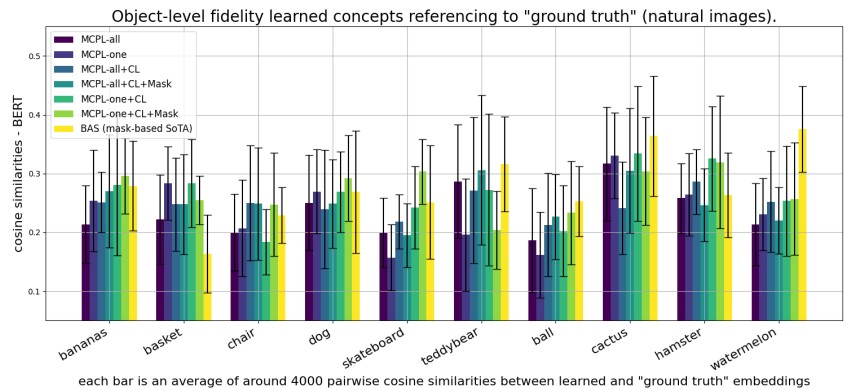

Figure 16:
footnotesize Per-object (natural images) embedding similarity (BERT) of learned concept comparing to the masked "ground truth". We evaluate the embedding similarity of our multi-concept adaptation of Textural Inversion (MCPL-all) and the state-of-the-art (SoTA) mask-based learning method, Break-A-Scene (BAS) by Avrahami et al. (2023), against our regularised versions.

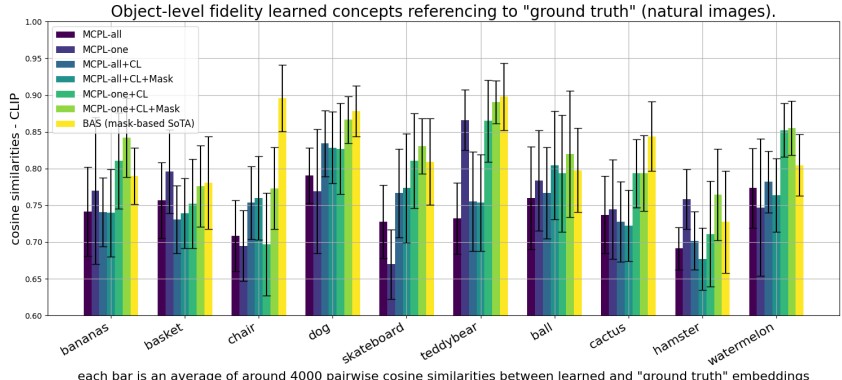

Figure 17: Per-object (natural images) embedding similarity (CLIP) of learned concept comparing to the masked "ground truth". We evaluate the embedding similarity of our multi-concept adaptation of Textural Inversion (MCPL-all) and the state-of-the-art (SoTA) mask-based learning method, Break-A-Scene (BAS) by Avrahami et al. (2023), against our regularised versions.

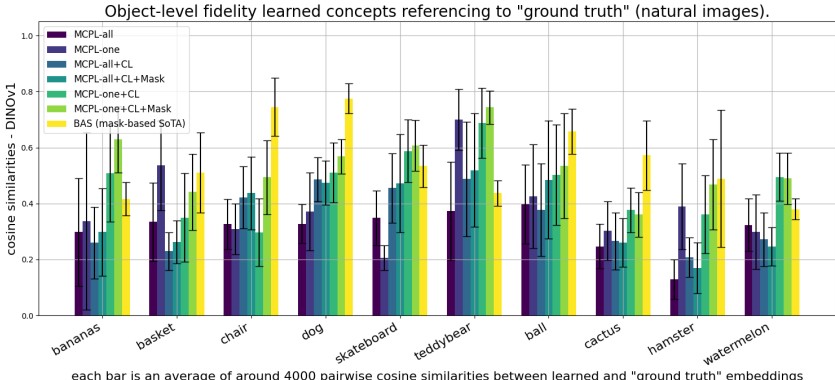

Figure 18: Per-object (natural images) embedding similarity (DINOv1) of learned concept comparing to the masked "ground truth". We evaluate the embedding similarity of our multi-concept adaptation of Textural Inversion (MCPL-all) and the state-of-the-art (SoTA) mask-based learning method, Break-A-Scene (BAS) by Avrahami et al. (2023), against our regularised versions.

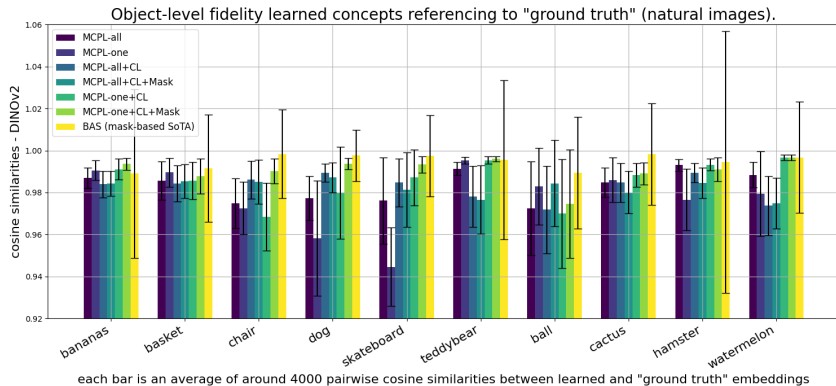

Figure 19: Per-object (natural images) embedding similarity (DINOv2) of learned concept comparing to the masked "ground truth". We evaluate the embedding similarity of our multi-concept adaptation of Textural Inversion (MCPL-all) and the state-of-the-art (SoTA) mask-based learning method, Break-A-Scene (BAS) by Avrahami et al. (2023), against our regularised versions.

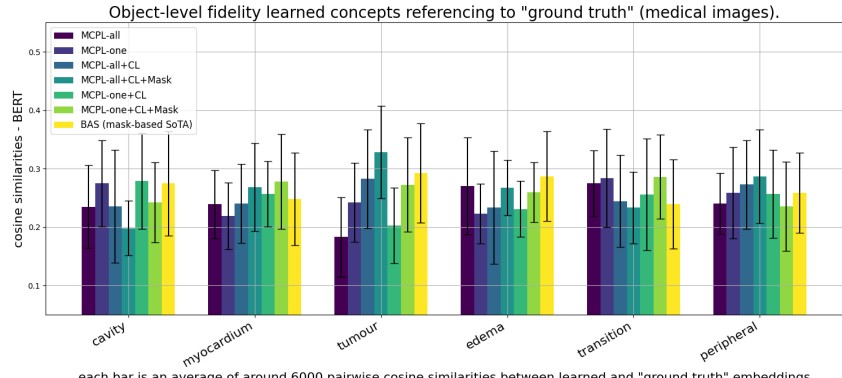

Figure 20: Per-object (medical images) embedding similarity (BERT) of learned concept comparing to the masked "ground truth". We evaluate the embedding similarity of our multi-concept adaptation of Textural Inversion (MCPL-all) and the state-of-the-art (SoTA) mask-based learning method, Break-A-Scene (BAS) by Avrahami et al. (2023), against our regularised versions.

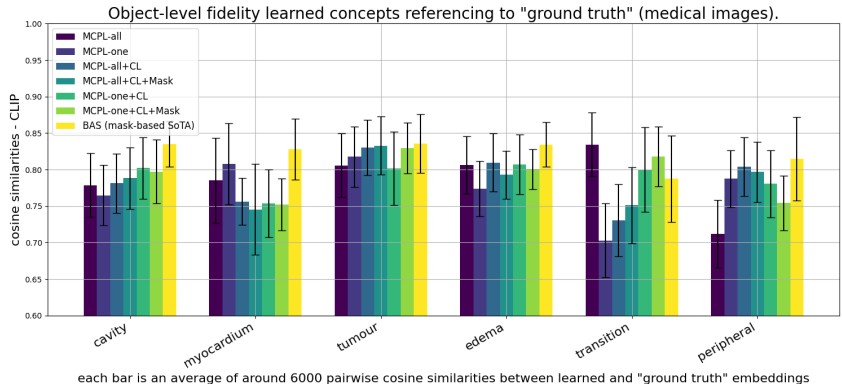

Figure 21: Per-object (medical images) embedding similarity (CLIP) of learned concept comparing to the masked "ground truth". We evaluate the embedding similarity of our multi-concept adaptation of Textural Inversion (MCPL-all) and the state-of-the-art (SoTA) mask-based learning method, Break-A-Scene (BAS) by Avrahami et al. (2023), against our regularised versions.

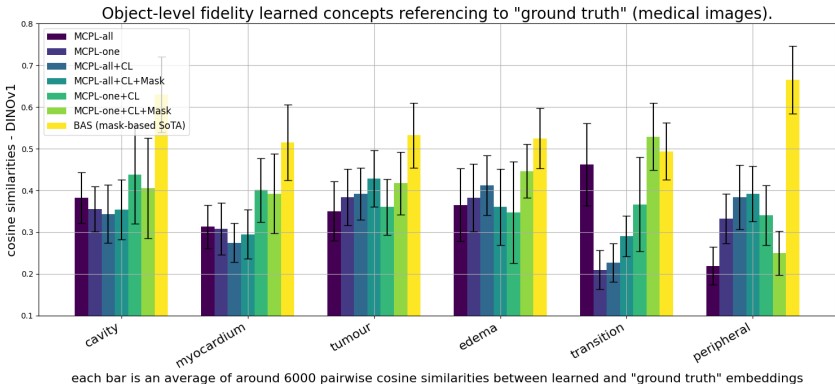

Figure 22: Per-object (medical images) embedding similarity (DINOv1) of learned concept comparing to the masked "ground truth". We evaluate the embedding similarity of our multi-concept adaptation of Textural Inversion (MCPL-all) and the state-of-the-art (SoTA) mask-based learning method, Break-A-Scene (BAS) by Avrahami et al. (2023), against our regularised versions.

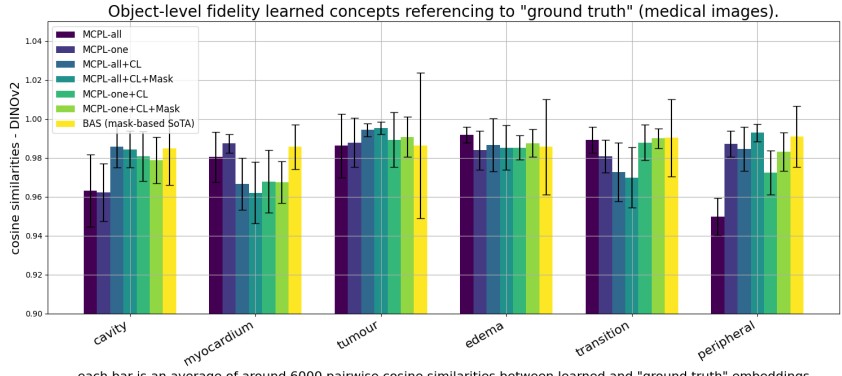

Figure 23: Per-object (medical images) embedding similarity (DINOv2) of learned concept comparing to the masked "ground truth". We evaluate the embedding similarity of our multi-concept adaptation of Textural Inversion (MCPL-all) and the state-of-the-art (SoTA) mask-based learning method, Break-A-Scene (BAS) by Avrahami et al. (2023), against our regularised versions.

## A.6 VISUALISATION OF ALL BREAK-A-SCENE RESULTS

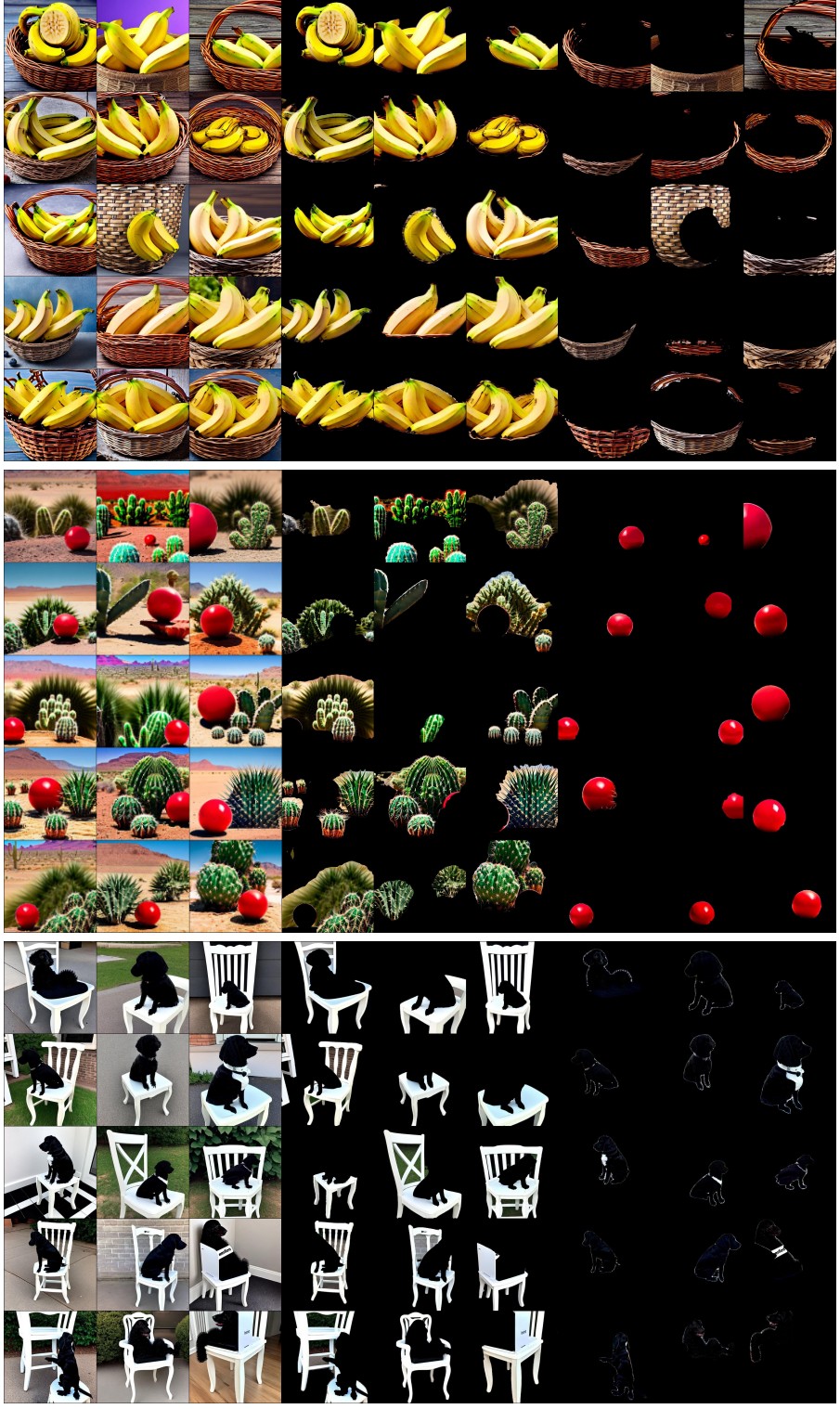

Figure 24: Visualisation of the Break-A-Scene results of generated and masked natural images.

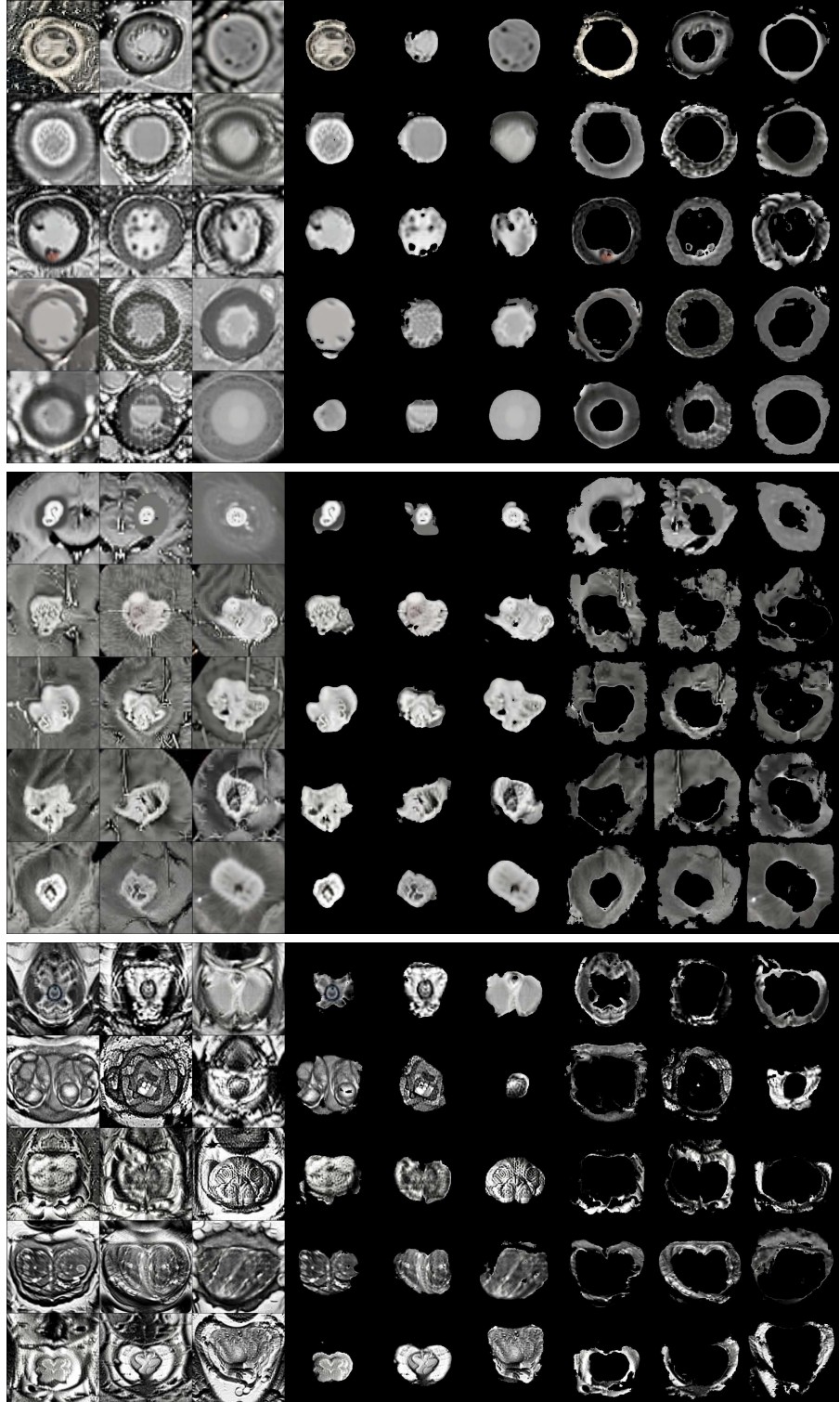

Figure 25: Visualisation of the Break-A-Scene results of generated and masked medical images.

## A.7 FULL T-SNE RESULTS

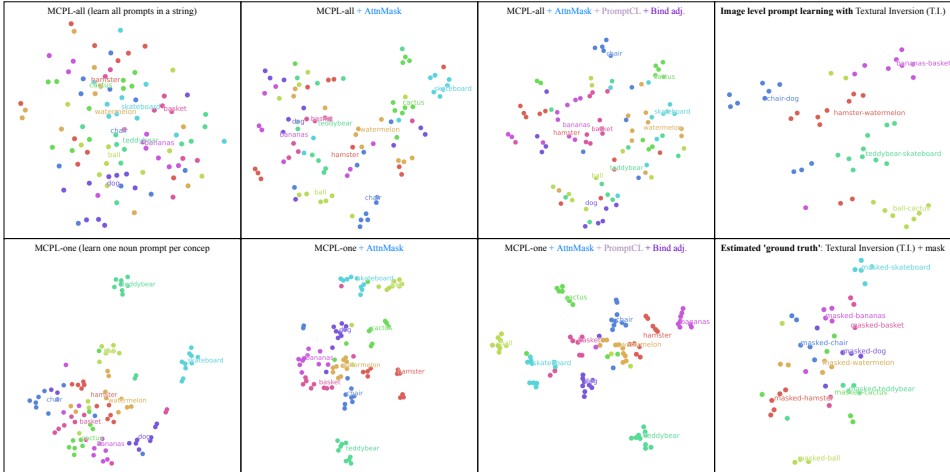

Figure 26: **The t-SNE visualisations of learned prompt-concept features (comparing all variants) on the in-distribution natural dataset**. We use features learned with *Textural Inversion Gal et al. (2022) on either unmasked or per-concept masked images*. We use the features learned with *Textural Inversion Gal et al. (2022) on per-concept masked images* to approximate the unknown 'Ground truth'. We compare two versions of our proposed MCPL framework *MCPL-all (learn all prompts in a string* and *MCPL-ong (learn one noun prompt per concept*. We also compare two variants one each version of our method, where we add the proposed regularisation of *AttnMask* or *AttnMask + PromptCL + Bind adj.* over the multi-concepts learning goal.

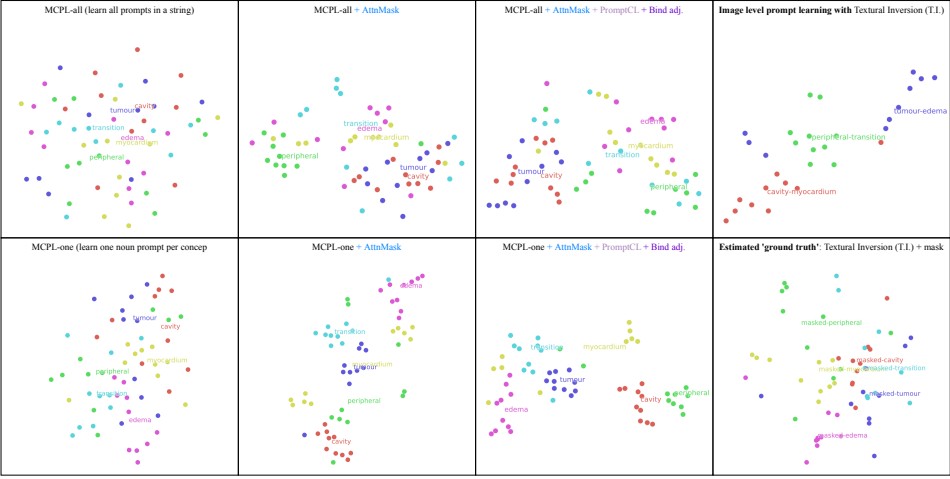

Figure 27: **The t-SNE visualisations of learned prompt-concept features (comparing all variants) on the out-distribution medical dataset**. We use features learned with *Textural Inversion Gal et al. (2022) on either unmasked or per-concept masked images*. We use the features learned with *Textural Inversion Gal et al. (2022) on per-concept masked images* to approximate the unknown 'Ground truth'. We compare two versions of our proposed MCPL framework *MCPL-all (learn all prompts in a string* and *MCPL-ong (learn one noun prompt per concept*. We also compare two variants one each version of our method, where we add the proposed regularisation of *AttnMask* or *AttnMask + PromptCL + Bind adj.* over the multi-concepts learning goal.

## A.8  MORE QUALITATIVE RESULTS

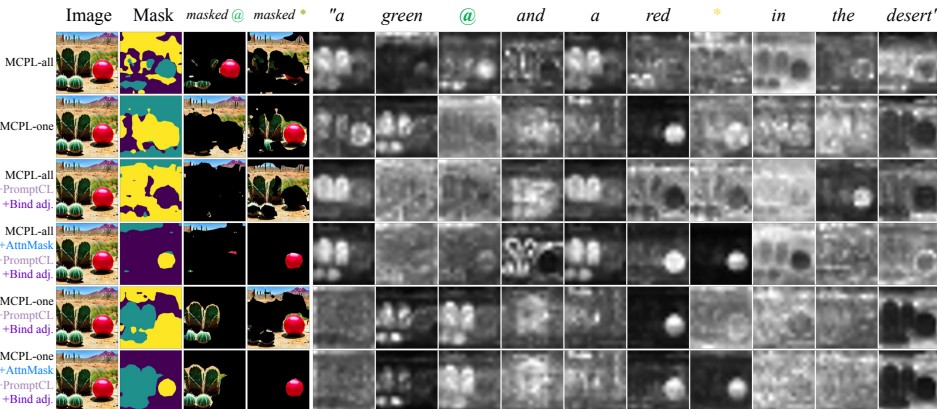

Figure 28: Visualisation of generated in-distribution natural concepts @ ("cactus") and * ("ball") with ours and all baseline methods.

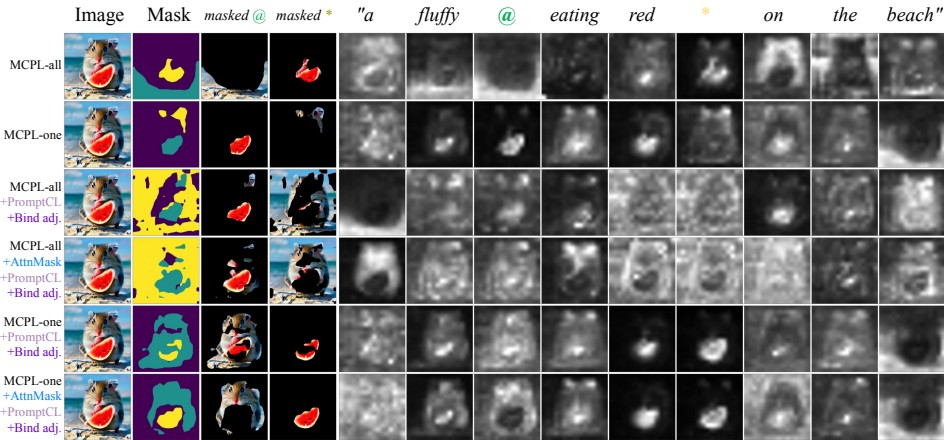

Figure 29: Visualisation of generated in-distribution natural concepts @ ("hamster") and * ("watermelon") with ours and all baseline methods.

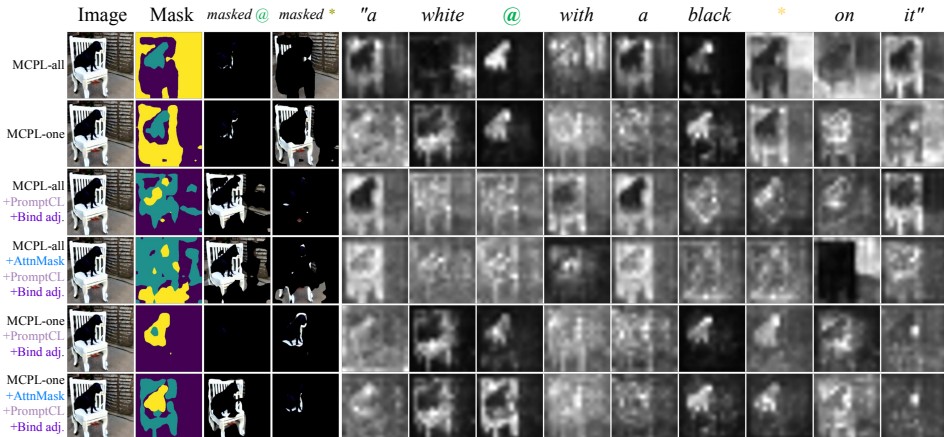

Figure 30: Visualisation of generated in-distribution natural concepts @ ("chair") and * ("dog") with ours and all baseline methods.

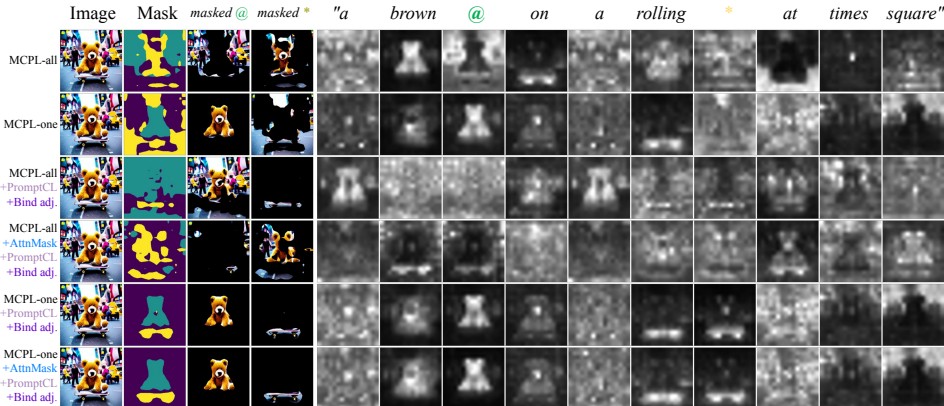

Figure 31: Visualisation of generated in-distribution natural concepts @ ("teddybear") and * ("skateboard") with ours and all baseline methods.

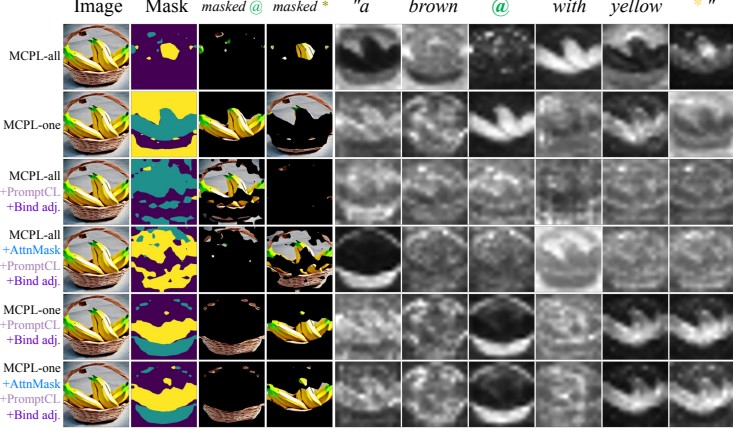

Figure 32: Visualisation of generated in-distribution natural concepts @ ("basket") and * ("bananas") with ours and all baseline methods.

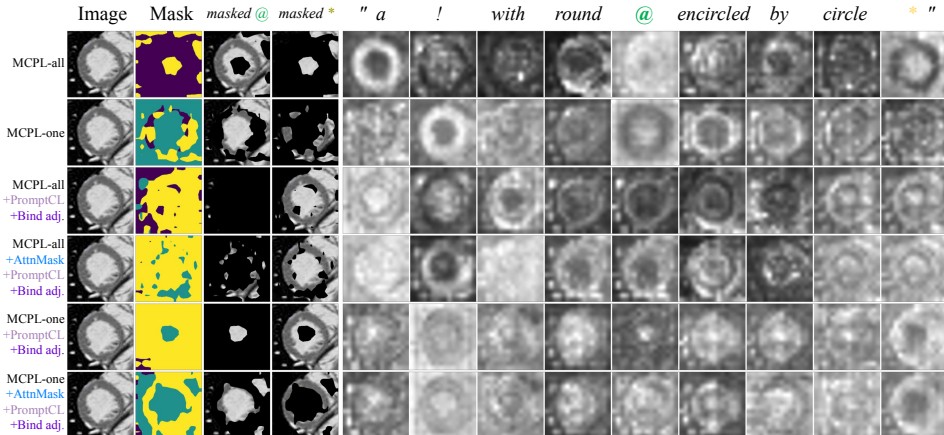

Figure 33: Visualisation of generated out-of-distribution medical concepts @ ("cavity") and * ("myocardium") with ours and all baseline methods.

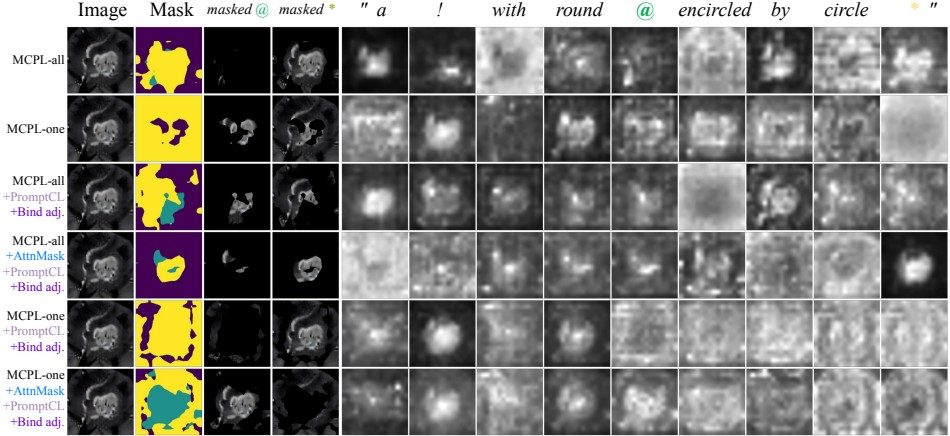

Figure 34: Visualisation of generated out-of-distribution medical concepts @ ("tumour") and * ("edema") with ours and all baseline methods.

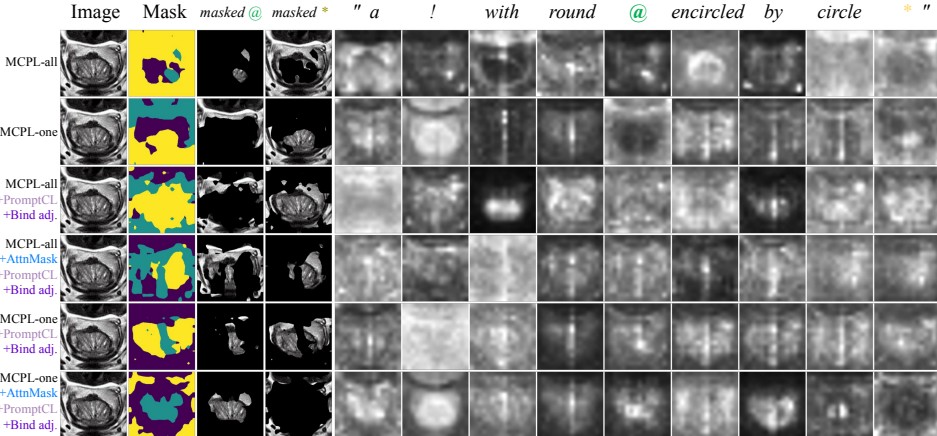

Figure 35: Visualisation of generated out-of-distribution medical concepts @ ("transition") and * ("peripheral") with ours and all baseline methods.

## A.9 FULL MOTIVATIONAL EXPERIMENT RESULTS

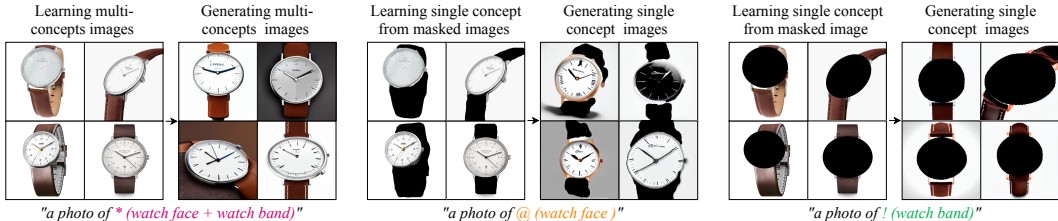

Figure 36: **Motivational study with watch images.** We learn embeddings using Textural Inversion on both unmasked multi-concept images ("watch face" and "watch band") and masked single-concept images ("watch face" or "watch band").

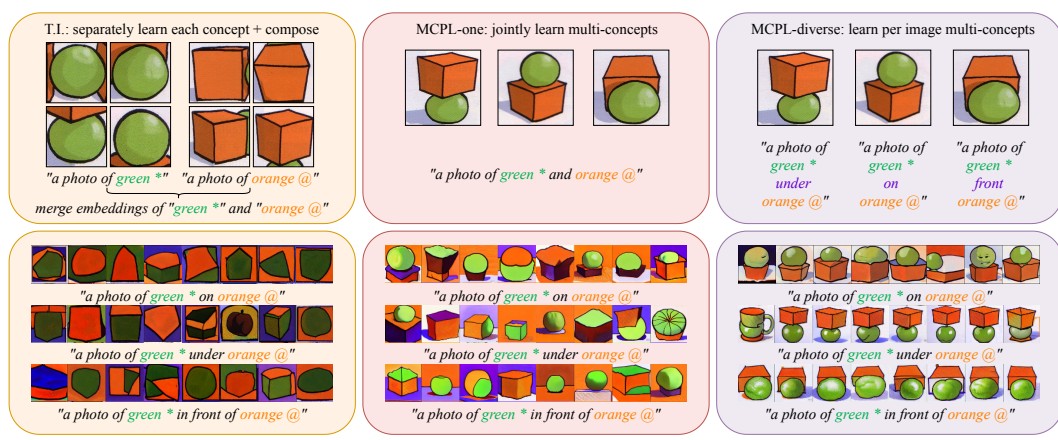

Figure 37: **Learning and Composing "ball" and "box"**. We learned the concepts of "ball" and "box" using different methods (top row) and composed them into unified scenes (bottom row). We compare three learning methods: *Textural Inversion* (Gal et al., 2022), which learns each concept separately from isolated images (left); *MCPL-one*, which jointly learns both concepts from uncropped examples using a single prompt string (middle); and *MCPL-diverse*, which advances this by learning both concepts with per-image specific relationships (right).

## A.10 FULL ABLATION RESULTS OF ASSESSING REGULARISATION TERMS WITH CROSS-ATTENTION

We present in this section the full results of assessing our proposed regularisation terms in Section 3.4. The results presented in Figure 38 indicate that our plain *MCPL* algorithm can learn complex multi-object relationships, yet it may not accurately capture semantic correlations between prompts and objects. To address this, we introduce several regularisation terms: *AttnMask*, *PromptCL*, and *Bind adj.*. We assess the efficacy of these terms in disentangling learned concepts by visualising attention and segmentation masks, as shown in Figure 38. Although the primary aim of this work is not segmentation accuracy, we generate segmentation masks of target concepts to provide a visual quantification of disentanglement. Our visual results suggest that incrementally incorporating the proposed regularisation terms enhances concept disentanglement, whereas applying them in isolation yields suboptimal outcomes. Moreover, the results demonstrate that *MCPL-one* is a more effective learning strategy than *MCPL-all*, highlighting the importance of excluding irrelevant prompts to maintain a focused learning objective.

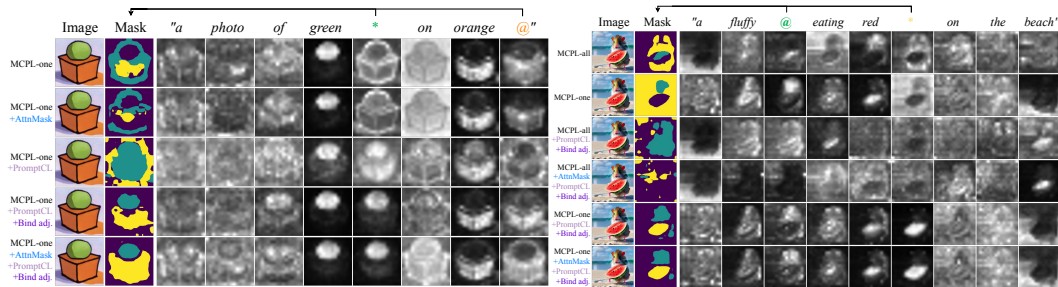

Figure 38: **Enhancing object-level prompt-concept correlation in MCPL using proposed *Attn-Mask*, *PromptCL* and *Bind adj.* regularisation techniques.** We conduct ablation studies to assess the impact of our proposed regularisation methods. We apply these techniques to the *MCPL-one* framework, using a "Ball and Box" example (left) and compare the performance of *MCPL-one* against *MCPL-all* in a "Hamster and Watermelon" example (right). We use average cross-attention maps to quantify the correlation of each prompt with its corresponding object-level concept. Additionally, we construct attention-based masks from multiple selected prompts for the concepts of interest.

### A.11 ALGORITHMS OF TEXTURAL-INVERSION AND MCPL

To introduce diversity, during training the generation is conditioned on phrases constructed from randomly selected text template $y$ derived from CLIP ImageNet (Radford et al., 2021) and the new prompt $p^*$, such as "A photo of $p^*$", "A sketch of $p^*$", etc.

---

**Algorithm 1:** Textural-Inversion

1   **Input:** a small set of images $x_0$, pre-trained text-encoder $c_\theta$ and denoising network $\epsilon_\theta$.
2   **Output:** an embedding $v^*$ corresponds to new prompt $p^*$.
3   initialise $v^* = c_\theta(p^*)$ ;
4   # optimising $v^*$ with $L_{DM}$
5   **for** $step = 1, \ldots, S$ **do**
6      randomly sample neutral texts $y$ to make string $[y, p^*]$;
7      **for** $t = T, T-1, \ldots, 1$ **do**
8          $v^* := \arg\min_v E_{x_0, \epsilon \sim N(0,I)}$
9          $\|\epsilon - \epsilon_\theta(x_t, t, [c_\theta(y), v^*])\|^2$;
10     **end**
11   **end**
12   **Return** $(p^*, v^*)$

---

### A.12 DATASET PREPARATION.

For the in-distribution natural images dataset, we generate variations of target objects using local text-driven editing, as proposed by Patashnik et al. (2023). This minimizes the influence of irrelevant elements like background. This approach also produces per-text local masks based on attention maps, assisting us in getting our best approximation for the "ground truth" of disentangled embeddings. We generate five sets of natural images containing 10 object-level concepts. For the out-of-distribution bio-medical image dataset, we assemble three sets of radiological images featuring six organ/lesion concepts. These images are sourced from three public MRI segmentation datasets: heart myocardial infarction (Lalande et al., 2020), prostate segmentation (Antonelli et al., 2022), and Brain Tumor Segmentation (BraTS) (Menze et al., 2014). Each dataset includes per-concept masks. For both natural and biomedical datasets, we collected 40 images for each concept. Figure 39 gives some examples of the prepared datasets.

---

**Algorithm 2:** MCPL

1  **Input:** a small set of images $x_0$, pre-trained text-encoder $c_\theta$ and denoising network $\epsilon_\theta$.
2  **Output:** a list of multiple embeddings $\mathcal{V} = [v^*, \dots, v^\&]$ corresponds to multiple new prompts $\mathcal{P} = [p^*, \dots, p^\&]$.
3  initialise $[v^*, \dots, v^\&] = [c_\theta(p^*), \dots, c_\theta(p^\&)]$ ;
4  # optimising $\{v^*, \dots, v^\&\}$ with $L_{DM}$
5  **for** $step = 1, \dots, S$ **do**
6      randomly sample neutral texts $y$ to make string $[y, p^*, \dots, p^\&]$;
7      **for** $t = T, T-1, \dots, 1$ **do**
8          $[v^*, \dots, v^\&] := \arg\min_\mathcal{V} E_{x_0, \epsilon \sim N(0, I)}$
9          $\|\epsilon - \epsilon_\theta(x_t, t, [c_\theta(y), v^*, \dots, v^\&]\|^2$;
10     **end**
11 **end**
12 **Return** $(\mathcal{P}, \mathcal{V})$

---

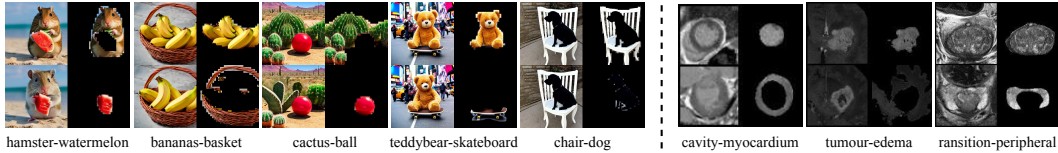

Figure 39: **Quantitative evaluation dataset examples.** We prepared five sets of in-distribution natural images and three sets of out-of-distribution biomedical images, each containing two concepts resulting in a total of 16 concepts. Visualisation of full sets is available in the Appendix A.13.

## A.13 VISUALISATION OF QUANTITATIVE EVALUATION DATASETS

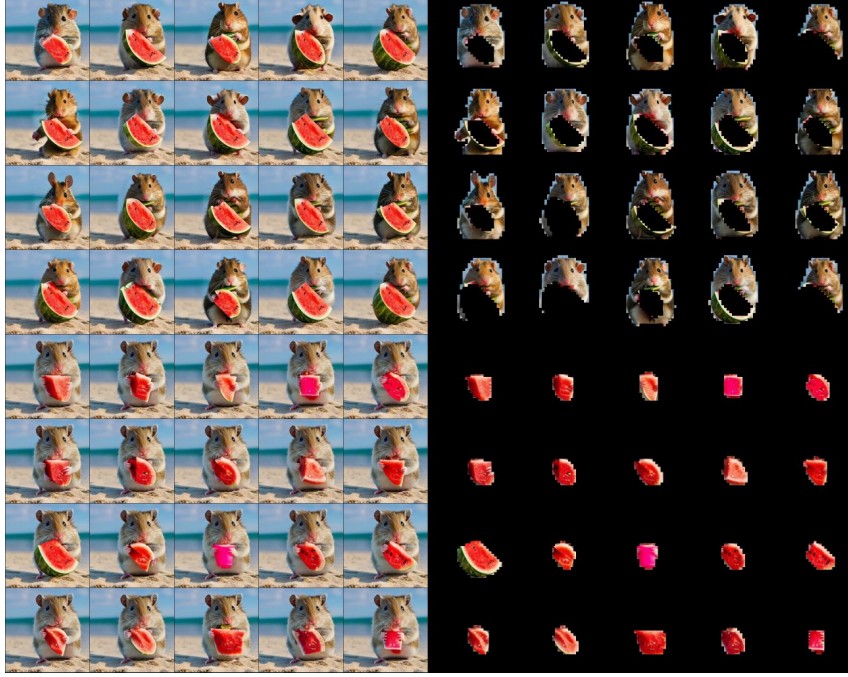

Figure 40: Visualisation of the full sets of generated and masked hamster-watermelon images.

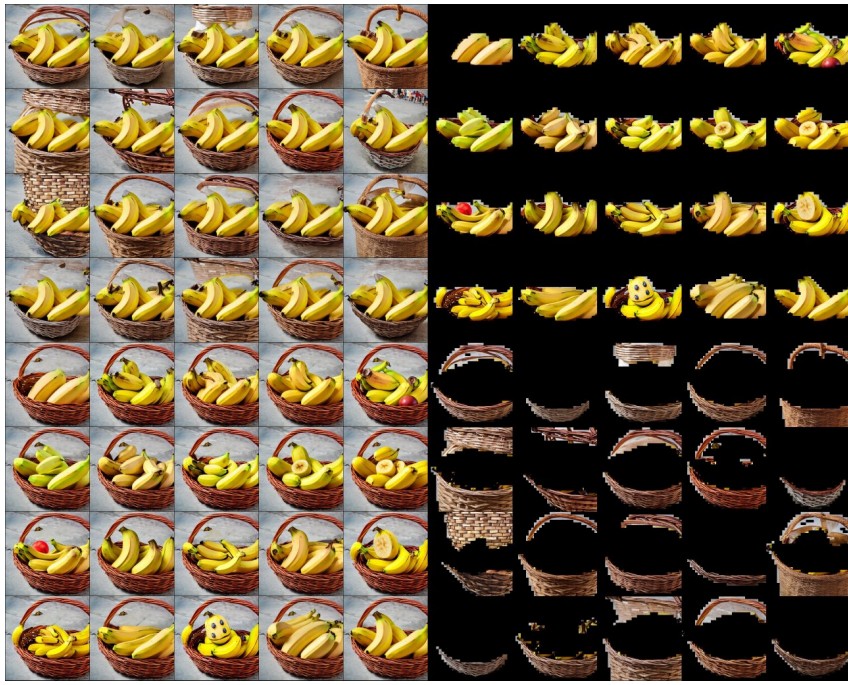

Figure 41: Visualisation of the full sets of generated and masked bananas-basket images.

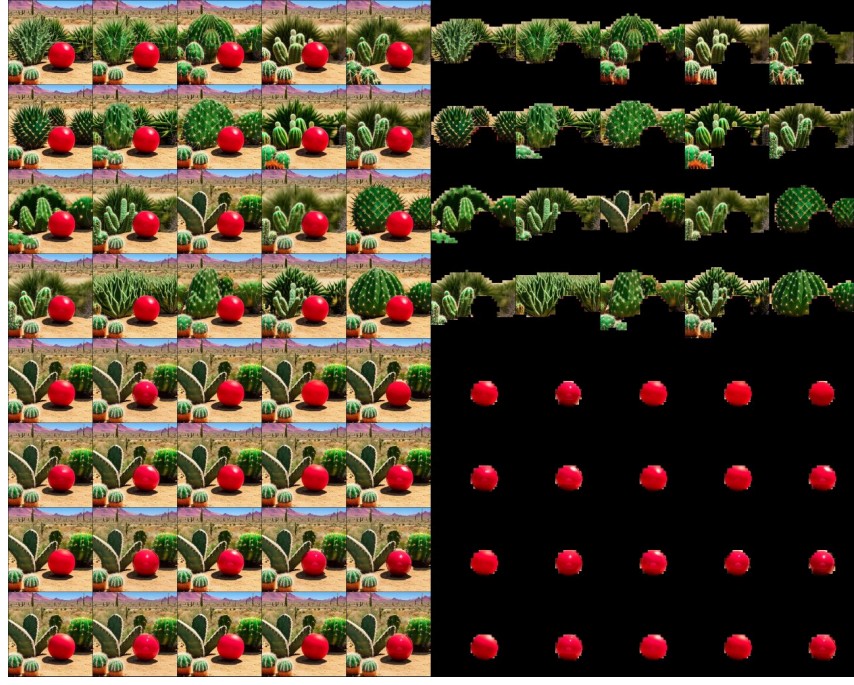

Figure 42: Visualisation of the full sets of generated and masked cactus-ball images.

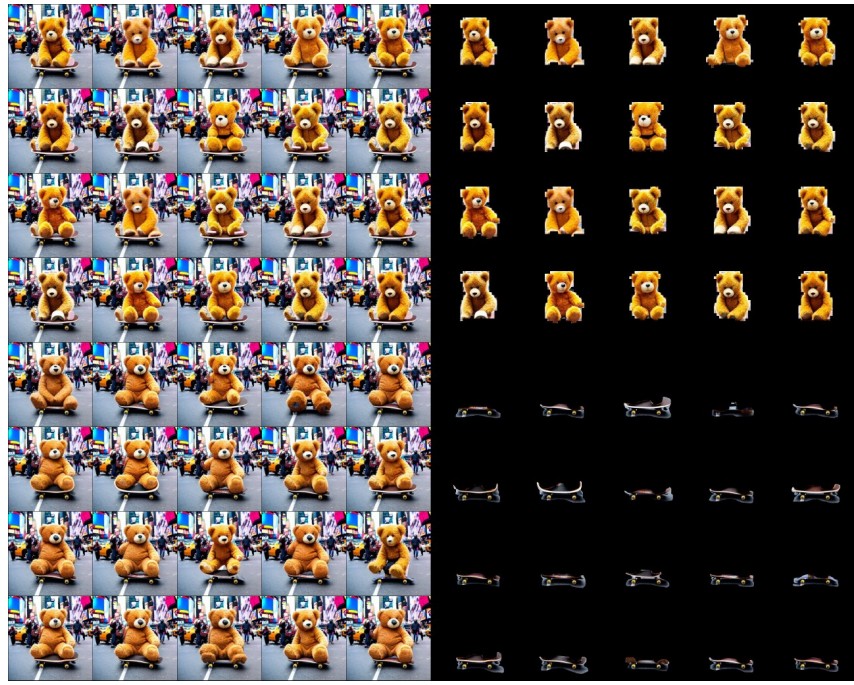

Figure 43: Visualisation of the full sets of generated and masked teddybear-skateboard images.

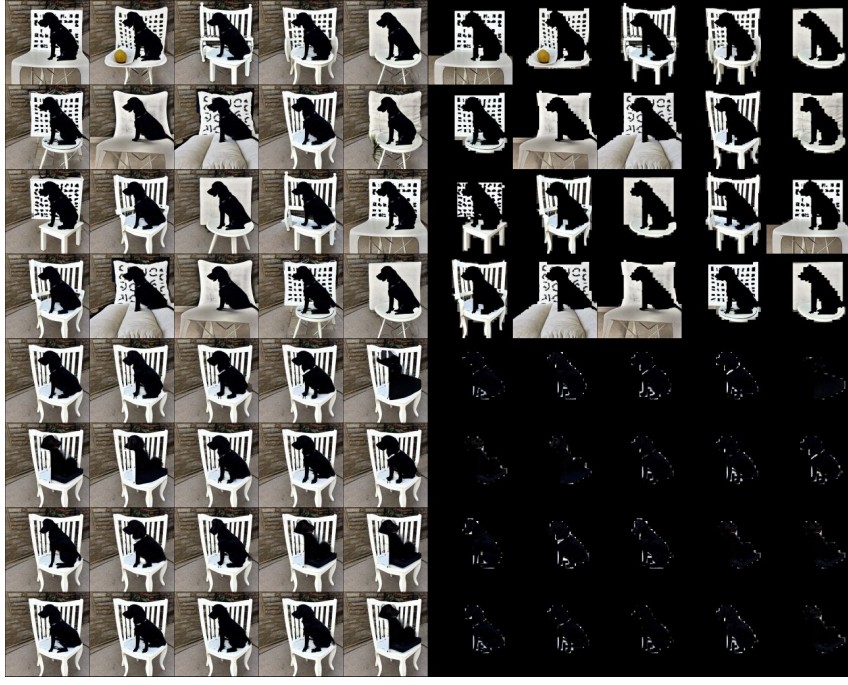

Figure 44: Visualisation of the full sets of generated and masked chair-dog images.

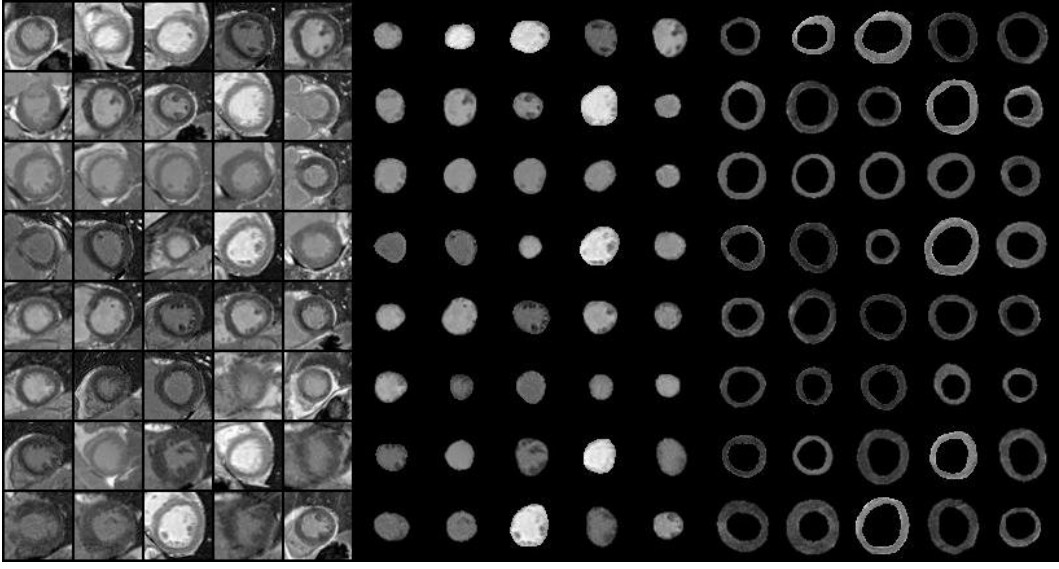

Figure 45: Visualisation of the full sets of generated and masked cavity-myocardium images.

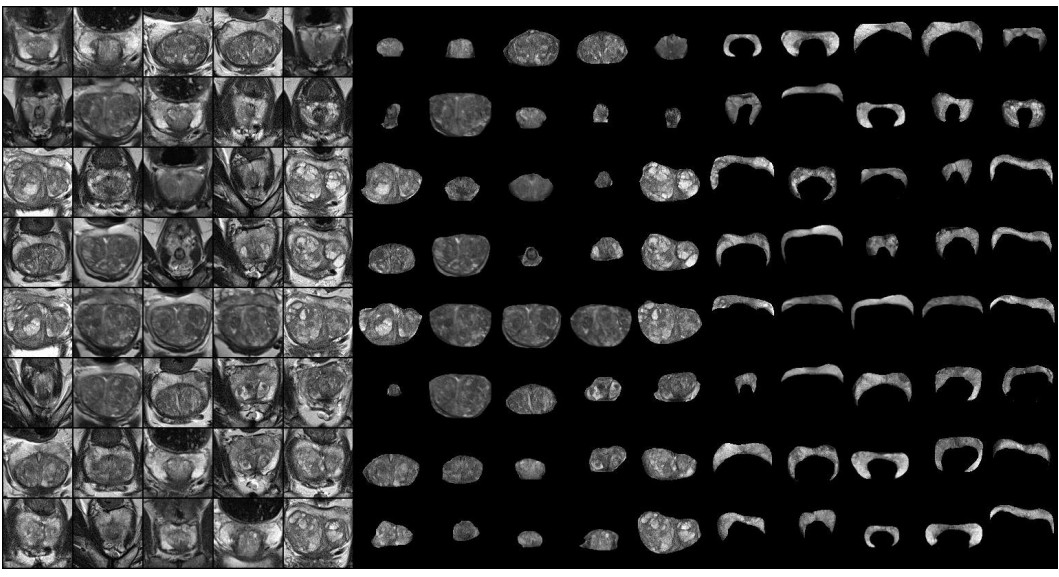

Figure 46: Visualisation of the full sets of generated and masked transition-peripheral images.

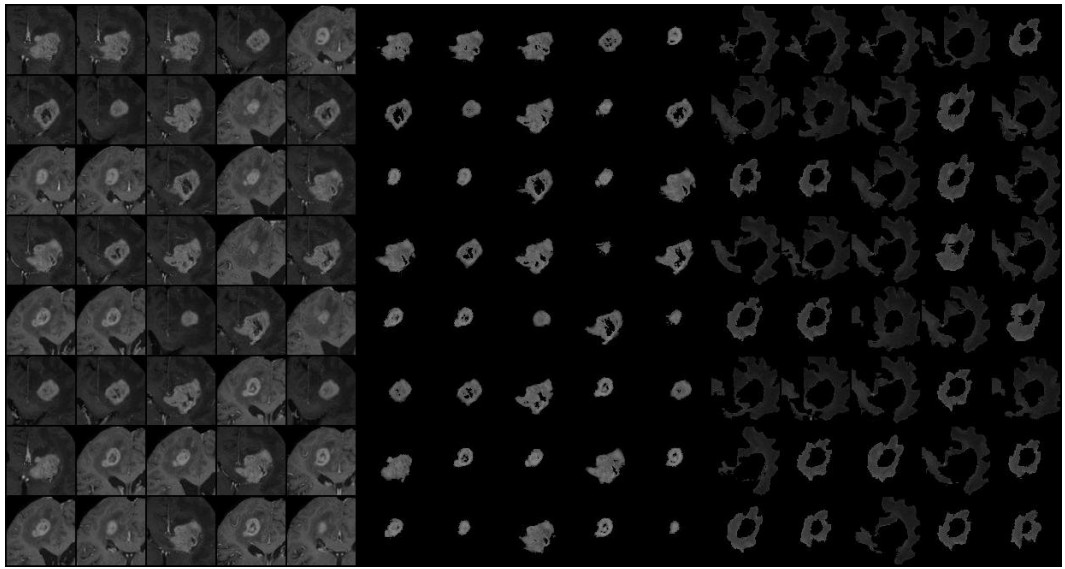

Figure 47: Visualisation of the full sets of generated and masked tumour-edema images.

