# OpenReview forum: "An Image is Worth Multiple Words: Learning Object Level Concepts using Multi-Concepts Prompts Learning"
_ICLR.cc/2024/Conference — Submitted to ICLR 2024_

### Official Review · Reviewer_wHAP · 2023-10-25

**Soundness:** 3 good
**Presentation:** 4 excellent
**Contribution:** 3 good
**Rating:** 6
**Confidence:** 3

**Summary:**

This paper introduces the Multi-Concept Prompt Learning (MCPL) framework for simultaneously learning multiple prompts from one scene in order to address the challenge of managing multiple concepts in scenes with multiple objects. The authors conducted a motivational study to investigate the limitations of existing prompt learning methods in multi-concept settings and found that object-level learning and editing without manual intervention remains challenging. To enhance prompt-object level correlation, the authors propose regularization techniques including Attention Masking (AttnMask) and Prompts Contrastive Loss (PromptCL). Experimental results demonstrate that the MCPL framework enables enhanced precision in object-level concept learning, synthesis, editing, quantification, and understanding of relationships between multiple objects.

**Strengths:**

1.	A novel task of the Multi-Concept Prompt Learning (MCPL) framework, which enables simultaneous learning of multiple prompts from one scene. This approach addresses the challenge of learning multiple concepts within multi-object scenes, which has not been previously explored.

2.	Enhanced Object-Level Concept Learning: The proposed MCPL framework demonstrates enhanced precision in object-level concept learning, synthesis, editing, quantification, and understanding of relationships between multiple objects. This is validated through extensive quantitative analysis and evaluation of learned object-level embeddings.

3.	The paper proposes several regularization techniques to enhance the accuracy of prompt-object level correlation. These techniques restrict prompt learning to relevant regions, facilitate disentanglement of prompt embeddings, and the use of descriptive adjective words to bind each learnable prompt. These effective techniques contribute to learning object-level information under image-level supervision.

**Weaknesses:**

1.	The aim of this paper is to learn and compose multiple concepts in the same scene. However, the demonstrations to prove the composing ability are insufficient. In almost all demonstrations the concepts are composed in the same string as training examples without any changes and only some editing demonstrations are available. Upon the interaction between the two concepts is changed, the effects seem to be worse.

2.	Some writing mistakes exist in the paper. In the top right of Figure 3, the labeling of “on” and “under” should be reversed. On page 6, “The” in “Therefore The contrastive loss” should be lowercase in line 8 of the paragraph before “Implementation details”.

3.	The experiment is a little confusing. In the section “Baselines and experiments” on page 7, the author presents four learning methods to compare their effectiveness. However, the author seems not to explain the meaning of the first setting called textural Inversion applied to unmasked multi-concept images and the subsequent experiments don’t contain the effects of the first two settings. The detail and comparison for a setting called “MCPL-diverse” are also unavailable.

4.	The paper constructs a new dataset to evaluate the proposed framework for multi-concept learning. However, I see all provided examples contain only two distinct concepts so I doubt the generalization of this method.

**Questions:**

Please see the weakness.

---

> ### Author Response · Authors · 2023-11-23
> **Respond to R4 (wHAP)**
>
> We are pleased to note that the reviewer **recognises the novel task** of learning multiple concepts within multi-object scenes under image-level supervision, which has **not been previously explored** and the proposed Multi-Concept Prompt Learning (MCPL) framework in conjunction with proposed regularisation techniques **effectively tackles such challenge**. Specifically, we are gratified that the reviewer:
>
> * Confirmed our observations from the motivational study that **existing method has limitations in object-level learning and editing without manual intervention**;
> * Agreed that our **extensive quantitative analysis and evaluation** show that the proposed approaches effectively enhanced precision in object-level concept learning, synthesis, editing and quantification.
> * Acknowledged **our proposed regularisation terms effective techniques contribute to learning object-level information under image-level supervision2**;
>
> We are also thankful for the reviewer’s identification of certain weaknesses and questions in our work. We have taken these points into careful consideration, performed extensive additional experiments and provided clarifications as follows.
>
> ---
>
>  > _… the demonstrations to prove the composing ability are insufficient..upon the interaction between the two concepts is changed._
>
> > _… However, I see all provided examples contain only two distinct concepts so I doubt the generalization of this method._
>
> * We thank the reviewer for raising this important experiment. Both ‘composing capability’ and ‘learning more than two concepts' are covered in Figure 11-12 of Section A.2, which involves **learning three or more concepts from a single image and composing a subset in a new scene**.
> * **Both tasks involve composing with strings differ from training**, our method generally learns targeted concepts and showed decent composing capability, outperforming Textural Inversion and DreamBooth, and was closer to **Break-A-Scene**, which represents the **state-of-the-art and a performance upper-bound in masked-based multi-concept learning**. **Unlike BAS, our approach neither uses segmentation masks as input nor updates model parameters.**
>
> ---
>
>  > _… The detail and comparison for a setting called “MCPL-diverse” are also unavailable._
>
> * We are thankful for the reviewer’s question and interest in “MCPL-diverse” — a powerful but under-explored variant of our method in the original submission. We perform new experiments and provide clarifications as follows.
> * **MCPL-diverse v.s. MCPL-one with segmentation masks**: we add new qualitative results in Figures 13 and 14 in Section A.3 for the task of learning per-image different concepts. In both examples, as **MCPL-diverse is specially designed for such tasks, it consistently outperforms MCPL-one**.
> * MCPL-diverse has also been found powerful in more competitive tasks of learning more than three concepts from a single image, following R2(RxXV)’s suggestion on comparing competitive baseline Break-A-Scene, which represents the **state-of-the-art and a performance upper-bound in masked-based multi-concept learning**. **Unlike BAS, MCPL neither uses segmentation masks as input nor updates model parameters.**
> * While R2 noted it's somewhat **unfair to compare directly with BAS due to its segmentation inputs**, we add Figure 12 of Section A.2 for this ambitious comparison. In Figure 12, we evaluated both MCPL-one and MCPL-diverse, the latter employing random crops akin to BAS’s ’union sampling’ strategy. The random crops strategy leads to each image containing different multiple concepts. In such case, **we found MCPL-diverse is a perfect fit, outperforming MCPL-one and comparable to the ambitious BAS baseline.**
>
> ---
>
>  > _... not to explain the meaning of the first setting called textural Inversion applied to unmasked multi-concept images and the subsequent experiments don’t contain the effects of the first two settings._
>
> * We thank the reviewer for identifying this clarification issue! The first setting called ‘textural Inversion applied to unmasked multi-concept images was a very initial baseline we explored. It was designed to assess the learning of combined concepts (i.e. encoding two concepts in one prompt embedding). This baseline was evaluated by the t-SNE visualisation and because it was **less comparable to other experiments** hence has been presented in Figures 26 and 27 in Appendix A.5.
> * We apologise for the above confusion and revised the entire experiments section to further clarify all experiment setups.
>
> ---
>
>  > _Some writing mistakes exist in the paper. In the top right of Figure 3, the labelling of “on” and “under” should be reversed. On page 6, “The” in “Therefore The contrastive loss” should be lowercase in line 8 of the paragraph before “Implementation details”._
>
> * We thank the reviewer for the thoughtful reviews and for pointing out those writing mistakes, we updated Figure 3 and corrected the writing mistake in the revised submission.

---

### Official Review · Reviewer_cRiH · 2023-10-30

**Soundness:** 3 good
**Presentation:** 3 good
**Contribution:** 3 good
**Rating:** 8
**Confidence:** 3

**Summary:**

To tackle the issue of learning multiple individual concepts simltaneously, in this paper, the authors propose MCPL. The authors first conduct preliminary studies to show that vanilla MCPL to jointly learn multiple concepts is feasible, but it is not adequate to learn correlations between objects and locate corresponding concepts. To tackle this issue, the authors propose three techniques: AttnMask, PromptCL, and PromptCL with Bind adj.. The proposed full MCPL-one can correctly recognize and localize different concepts.

**Strengths:**

1. The proposed prompts contrastive loss as well as Bind adj. can effectively regularize the attention maps regarding concepts to localize onto correct position. And learning with AttnMask can also largely refine the attention mask boundary, thus reducing false positive attention values. All these methods benefit to textural inversion.

2. Extensive experimental results illustrate that the proposed MCPL can effectively generate attention masks for corresponding concepts, which benefits to textural inversion task.

3. The description of method and experiment section is polished and easy to understand.

**Weaknesses:**

The main concern is the analysis between MCPL-diverse mentioned in preliminary study and the full version of MCPL-one. The authors could provide visualization of generated natural concepts from MCPL-diverse as well as corresponding generated segmentation masks to support the observation.

**Questions:**

N/A

---

> ### Author Response · Authors · 2023-11-23
> **Respond to R3 (cRiH)**
>
> We are pleased to note that the reviewer **recognises the challenge of learning multiple individual concepts simultaneously from a single image** and finds our work to be **well-motivated with detailed results**. Specifically, we are gratified that the reviewer:
>
> * Confirmed our proposed **three regularisation techniques**, Attention Masking, Prompts Contrastive Loss, and Bind Adjectives, **are effective and would benefit relevant works such as textural inversion**;
> * Agreed that our **extensive experiments** show that the proposed MCPL effectively creates attention masks for relevant concepts.
> * Acknowledged **the description of the method and experiment section is polished and easy to understand**;
>
> We are also thankful for the reviewer’s question and interest in **“MCPL-diverse” — a powerful but under-explored variation of our method in the original submission**. We perform new experiments and provide clarifications as follows
>
>  > _… analysis between the MCPL-diverse mentioned in the preliminary study and the full version of MCPL-one. The authors could provide visualization of generated natural concepts from MCPL-diverse as well as corresponding generated segmentation masks to support the observation._
>
> * **MCPL-diverse v.s. MCPL-one with segmentation masks**: we add new qualitative results in Figures 13 and 14 in Section A.3 for the task of learning per-image different concepts. In both examples, as **MCPL-diverse is specially designed for such tasks, it consistently outperforms MCPL-one**.
> * MCPL-diverse has also been found powerful in more competitive tasks of learning more than three concepts from a single image, following **R2(RxXV)**’s suggestion on comparing competitive baseline **Break-A-Scene**, which represents the **state-of-the-art and a performance upper-bound in masked-based multi-concept learning**. **Unlike BAS, our MCPL neither uses segmentation masks as input nor updates model parameters.**
> * While R2 noted it's somewhat **unfair to compare directly with BAS due to its relying on segmentation inputs**, we add Figure 12 of Section A.2 for this ambitious comparison. In the right example of Figure 12, we evaluated both MCPL-one and MCPL-diverse, the latter employing random crops akin to BAS’s ’union sampling’ strategy. The random crops strategy leads to each image containing different multiple concepts. In such case, **we found MCPL-diverse is a perfect fit, outperforming MCPL-one and comparable to the ambitious BAS baseline.**

---

### Official Review · Reviewer_RxXV · 2023-10-31

**Soundness:** 3 good
**Presentation:** 4 excellent
**Contribution:** 3 good
**Rating:** 6
**Confidence:** 3

**Summary:**

The paper proposes a Multi-Concept Prompt Learning (MCPL) method that extracts multiple prompts from single images under the stable diffusion framework. A motivation study is first provided to demonstrate the current limitation. The proposed method is based on Textual Inversion and incorporates multiple regularisations (including Attention Masking, Prompts Contrastive Loss, and Bind Adjective) to disentangle multiple objects. The concepts are learned from a new dataset and evaluated by two designed protocols, followed by application visualizations.

**Strengths:**

-  To achieve multi-concept extractions from a single image, the proposed method leverages novel regularisation losses without relying on any groundtruth object segmentation
-  Overall, the paper is clearly structured and easy to follow. The motivational study introduces the problem and the current limitation in a systematic way
-  The experiments are well-designed with both real-world categories and out-of-domain biomedical images involved during the evaluation
-  Comprehensive analysis is conducted with t-SNE visualizations and embedding similarity evaluation.

**Weaknesses:**

- Some recent works (such as Break-A-Scene) on similar tasks could also be mentioned in the related work section. On the other hand, though it would be a bit unfair to directly compare with Break-A-Scene (due to its given segmentation inputs), it could still be interesting to treat its performance as an upper bound and comment on how a good segmentation mask would affect the learned concepts.
- More implementation details could be added, particularly on prompt initialization. It seems a bit unclear how to initialize all learnable embeddings by the same word, “photo” in a random manner. Besides, one may be curious about how the number of prompts is determined, especially for MCPL-all.
- It would be better to discuss the limitations of current work in the last section and point out some future improvement directions.

**Questions:**

- It seems that all learned concepts (nouns) are associated with a pre-defined adjective description in the prompt (such as “a green *”). Are these adjectives playing roles in disentangling the concepts? What if the initial prompt is provided in the form “A * and a @”?
- I wonder if there are any examples of cases when the number of objects in the image is mismatched with the number of learnable concepts

---

> ### Author Response · Authors · 2023-11-23
> **Respond to R2 (RxXV) --- Part 1**
>
> We are pleased to note that the reviewer **recognises the challenge of multi-concept extractions from a single image** and finds our work to be **well-motivated with detailed results**. Specifically, we are gratified that the reviewer:
>
> * Confirmed our method **leverages novel regularisation losses without relying on any groundtruth object segmentation**;
> * Agreed that our **paper is clearly structured and easy to follow**. The **motivational study introduces the problem and the current limitation in a systematic way**.
> * Acknowledged that **experiments are well-designed and analysis are comprehensive**;
>
> We are also thankful for the reviewer’s identification of certain weaknesses and questions in our work. We have taken these points into careful consideration, performed extensive additional experiments and provided clarifications as follows.
>
> ---
>
>  > _W.1 - “Some recent works (such as Break-A-Scene) on similar tasks could also be mentioned in the related work section. On the other hand, though it would be a bit unfair to directly compare with Break-A-Scene (due to its given segmentation inputs), it could still be interesting to treat its performance as an upper bound and comment on how a good segmentation mask would affect the learned concepts.”_
>
> * We wholeheartedly concur and are thankful for the reviewer highlighting this significant recent reference, which had previously been overlooked in our work. In response, we have **conducted extensive experiments to incorporate Break-A-Scene into all our quantitative analyses and visual comparisons**. This inclusion enriches our study and provides a more comprehensive understanding of our method in the context of current advancements
> * Specifically, we have updated Figures 5, 7, 8, and 16-23 to compare against **Break-A-Scene as a performance upper bound** in our 16 concepts experiments. **This expansion includes an additional 160 single GPU runs, approximately 2 million pairwise similarities, and extensive human-in-the-loop pre- and post-processing for integration with Break-A-Scene**, as detailed in Section A.1. Moreover, Figures 11-12 in Section A.2 provide a **visual comparison with Break-A-Scene as well as other baselines**.
> * It is worth noting **BAS requires segmentation masks as input and employs separate segmentation models to produce masked objects**, hence integration BAS involves extensive human-in-the-loop pre- and post-processing efforts, as detailed in Section A.1. In contrast, **our method is mask-free at learning and employs its own AttnMask to generate masked objects.**
> * The following tables highlight new results in Figure 8, our fully regularised method (MCPL-one+CL+Mask) shows **competitive performance against BAS on natural images**. For the **out-of-domain (OOD) medical dataset, BAS leads in the DINOv1 space, but we're comparable in others**. This is **due to our less precise object masks compared to BAS's human-assisted MedSAM segmentation**, shown in Figures 5 and 6.
>
> **Object-level fidelity learned concepts referencing to "ground truth" (natural images)**
>
> | exp_names           | bert          | clip          | dinov1        | dinov2        |
> |---------------------|---------------|---------------|---------------|---------------|
> | MCPL-one+CL+Mask    | 0.271 ± 0.078 | 0.821 ± 0.050 | 0.534 ± 0.112 | 0.991 ± 0.006 |
> | BAS                 | 0.276 ± 0.078 | 0.823 ± 0.050 | 0.552 ± 0.097 | 0.995 ± 0.030 |
>
> **Object-level fidelity learned concepts referencing to "ground truth" (medical images)**
>
> | exp_names           | bert          | clip          | dinov1        | dinov2        |
> |---------------------|---------------|---------------|---------------|---------------|
> | MCPL-one+CL+Mask    | 0.262 ± 0.072 | 0.792 ± 0.037 | 0.407 ± 0.081 | 0.983 ± 0.009 |
> | BAS                 | 0.267 ± 0.079 | 0.823 ± 0.043 | 0.560 ± 0.080 | 0.987 ± 0.021 |
>
> * **The new t-SNE results in** Figure 7 demonstrate that the learned embeddings from both the mask-based ’ground truth’ and BAS show less disentanglement compared to ours, attributable to their lack of a specific disentanglement objective, such as the PromptCL loss in MCPL.
> * We've reflected on all the mentioned points in our revision, including ensuring proper citation of Break-A-Scene throughout the paper.

---

> ### Author Response · Authors · 2023-11-23
> **Respond to R2 (RxXV) --- Part 2**
>
> > _W.3 - “It would be better to discuss the limitations of current work in the last section and point out some future improvement directions.”_
>
> * We appreciate the reviewer's valuable suggestion! We've included a discussion on limitations and future work in the '**Limitation and Conclusion**' section, a copy of which is provided below for your convenience in reviewing.
> * We found our method to suffer from the following limitations:
>     * (1) **Imperfect Masking**: Our reliance on image-level text descriptions, instead of segmentation masks, grants flexibility in exploring unknown concepts but results in less precise object boundary optimization. Future research could use our AttnMask as an input prompt to segmentation models for mask refinement.
>     * (2) **Composition Capability**: MCPL’s composition strength is weaker than BAS, as MCPL doesn’t update model parameters, unlike BAS. Integrating MCPL with weight optimization methods like BAS or DreamBooth may enhance performance, albeit at higher computational costs, which is a potential direction for future work.
>     * (3) **Evaluation Metrics**: Current quantification methods in this field (e.g. TI, DB, CD, BAS, and P2P), predominantly rely on prompt/embedding similarity due to the absence of more effective quantification mechanisms without known ground truth. This indicates a need for developing better evaluation metrics in future research.
>     * (4) Our method **relies on adjectives** serving as textual descriptors (e.g., colour) to differentiate between multiple concepts. While human-machine interaction using purely linguistic descriptions is generally preferred, challenges arise when two concepts are very similar and lack distinct visual cues in the image. In such cases, our method may struggle, and Break-A-Scene currently offers the best solution.
>
> ---
>
> > _Q.2 - I wonder if there are any examples of cases when the number of objects in the image is mismatched with the number of learnable concepts_
>
> * We showcase examples in Figure 11-12 of Section A.2, which involves: 1) learning three concepts from a single image and composing a subset in a new scene. 2) more challenging, entails learning six concepts from one image and composing 1 to 3 concepts in a novel scene.
> * In all **“mismatched tasks”**, our method generally learns targeted concepts and showed decent composing capability, **outperforming Textural Inversion and DreamBooth, and was closer to BAS.**
> * We conclude the **'Bind Adjectives' regularisation directs our model to the correct location**. This is evidenced by the **performance diminishes when adjectives are removed**, as seen in the right example of Figure 12.
>
> ---
>
>  > _Q.1 - It seems that all learned concepts (nouns) are associated with a pre-defined adjective description in the prompt (such as “a green *”). Are these adjectives playing roles in disentangling the concepts? What if the initial prompt is provided in the form “A * and a @”?_
>
> * We add Figure 15 in section A.4 for an ablation study on the impact of adjective words. **Adjective words are crucial in linking each prompt to the correct region**; without them, the model may struggle for regional guidance, **akin to the role of masks in Break-A-Scene**.
> * Figure 38 highlights how each regularisation technique enhances object-level prompt-concept correlation in MCPL. The left example shows **reduced accuracy in attention and masks without Bind Adjectives**.
> * The aforementioned Figure 12-right results reinforce the significance of adjectives in maintaining model performance.
>
> ---
>
>  > _W.2 - More implementation details could be added, particularly on prompt initialization. It seems a bit unclear how to initialize all learnable embeddings by the same word, “photo” in a random manner. _
>
> * It’s a writing mistake, thank you for pointing it out! **To clarify, all learnable embeddings are initialised using the encoding of pseudowords like ‘*’ or '@', rather than random initialization.** We intended to convey that 'during training, sentences/phrases are constructed from randomly selected text templates y derived from CLIP ImageNet'. This has been corrected in the revised manuscript."
>
> ---
>
>  > _W.2 - Besides, one may be curious about how the number of prompts is determined, especially for MCPL-all._
>
> * For MCPL-all, we learn all prompts/words in the sentence.

---

### Official Review · Reviewer_BYgx · 2023-11-06

**Soundness:** 2 fair
**Presentation:** 2 fair
**Contribution:** 2 fair
**Rating:** 3
**Confidence:** 3

**Summary:**

This paper presents a method to associate new words into concepts. The proposes three techniques: attention masking, prompts contrastive loss, and bind adjective. The applications they adopted is the image synthesis / editing when replacing some of the original concepts in a sentence, with a different concept (i.e. image editing over disentangled concepts). They also claim to introduce a novel dataset for this application.

**Strengths:**

- They claimed to release code and dataset upon publication.
 - The paper targets important research areas.

**Weaknesses:**

- The paper is not very clear to read. The idea seems to be straightforward, but the description of the method is a bit ambiguous. I have to read multiple times to make sure I understand the method accurately.
 - In experiments, the authors show multiple interesting qualitative results. However, there are very little quantitative results, and it is very hard to compare with other methods and understand the contribution of this effort.

**Questions:**

Attention masks and contrastive loss on different concepts seems to be a widely used method. It would be great if the authors can explain a bit more about the novelty of their work.

---

> ### Author Response · Authors · 2023-11-23
> **Respond to R1 (BYgx) - Part 1**
>
> We are pleased to note the reviewer's agreement that our work **targets critical research areas** in text-guided learning of multiple prompts from a single scene **without ground-truth object segmentation**. We are also thankful for the confirmation of our **contribution in introducing a novel dataset**.
>
> Furthermore, we acknowledge and appreciate the **reviewer's questions regarding novelty, quantitative evaluation and clarity in our manuscript**. We recognise that these issues may stem from misunderstandings. To address these concerns, we have provided detailed responses to each question below, aiming to clarify any ambiguities and enhance the overall comprehensibility of our work.
>
> ---
>
>  > _Q.1 - Attention masks and contrastive loss on different concepts seem to be a widely used method. It would be great if the authors can explain a bit more about the novelty of their work._
>
> We appreciate the reviewer's inquiry and are pleased to clarify our contributions as follows:
>
> * Firstly, we highlight the consensus among all reviewers on **our work targets on important research areas** of **mask-free text-guided learning of multiple prompts from a single scene**.
> * We introducing Multi-Concept Prompt Learning (MCPL), **the first** to achieve multi-concept learning **without relying on object masks as input**. This novelty has been recognised by all the other reviewers.
> * Our approach **differs to the leading mask-based method**, Break-A-Scene (BAS) as follows:
>     * BAS learns multiple concepts from images paired with object-level masks but we don’t use a mask;
>     * BAS updates both textural embeddings and model weights but we don’t update model weights;
> * We also summarise **other key contributions** our work have made:
>     * The **motivational study** introduces the problem and the current limitation with vanilla MCPL. Addressing this, we propose **three innovative regularisation techniques**: Attention Masking, Prompts Contrastive Loss, and Bind Adjectives, to disentangle multiple object concepts and enhance the accuracy of object-level concept learning.
>     * **Extensive experiments and analysis** demonstrate superior performance with our method against competitive baselines, delivering results comparable to the leading mask-based method, Break-A-Scene. This **highlights the potential of our mask-free approach**. We also provide detailed visualisations to showcase the practical applications of our method.
>     * We have compiled a **new dataset** for comprehensive evaluation. This dataset involves both in-domain natural images and out-of-domain biomedical images, **setting up a new benchmark for the field**.
>     * Overall, our approach not only enhances current methodologies but also **paves the way for novel applications**, such as facilitating knowledge discovery through natural language-driven interactions between humans and machines.
> * All of the above clarifications and contributions have been reflected in our revised manuscript.

---

> ### Author Response · Authors · 2023-11-23
> **Respond to R1 (BYgx) - Part 2**
>
> > _W.2 - However, there are very little quantitative results, and it is very hard to compare with other methods and understand the contribution of this effort._
>
> * We are grateful to the reviewer for highlighting a potential limitation in our study. As our work is **pioneering** in the field of multi-concept learning **without ground-truth object segmentation**, we acknowledge the **scarcity of directly comparable baselines**.
> * **Our original experiments were extensive**, taking about 80 days on a single GPU, a fact recognised by other reviewers. However, following R2's advice, we **added Break-A-Scene (BAS) as an additional benchmark**. We perform **extensive new experiments resulting in an extra seven pages in the revised version.** We have updated Figures 5, 7, 8, 16-23, adding Figures 10-12, and detailing in Sections A.1 and A.2 for a comprehensive comparison. This involved 160 more single GPU runs and around 2 million pairwise comparisons.
> * This thorough approach ensures our approach is **quantitatively compared** against **three baselines** (Textural Inversion, MCPL, and BAS) and a **qualitative comparison** against an **additional five baselines** (Figure 11-12), comprehensively covering the scope of the field.
> * It worth noting **BAS requires segmentation masks as input and employs separate segmentation models to produce masked objects**, hence integration BAS involves extensive human-in-the-loop pre- and post-processing efforts, as detailed in Section A.1. In contrast, **our method is mask-free at learning and employed its own AttnMask to generate masked objects.**
> * In the following results (Figure 8 in table format), our fully regularised method (MCPL-one+CL+Mask) shows **competitive performance against BAS on natural images**. For the **out-of-domain (OOD) medical dataset, BAS leads in the DINOv1 space, but we're comparable in others**. This is **due to our less precise object masks compared to BAS's human-assisted MedSAM segmentation**, shown in Figures 5 and 6.
>
> **Object-level fidelity learned concepts referencing to "ground truth" (natural images)**
>
> | exp_names           | bert          | clip          | dinov1        | dinov2        |
> |---------------------|---------------|---------------|---------------|---------------|
> | MCPL-all            | 0.235 ± 0.076 | 0.742 ± 0.050 | 0.311 ± 0.115 | 0.983 ± 0.010 |
> | MCPL-one            | 0.235 ± 0.073 | 0.760 ± 0.065 | 0.388 ± 0.143 | 0.978 ± 0.014 |
> | MCPL-all+CL         | 0.247 ± 0.081 | 0.756 ± 0.051 | 0.346 ± 0.115 | 0.983 ± 0.010 |
> | MCPL-all+CL+Mask    | 0.251 ± 0.083 | 0.756 ± 0.057 | 0.362 ± 0.128 | 0.982 ± 0.012 |
> | MCPL-one+CL         | 0.265 ± 0.089 | 0.791 ± 0.061 | 0.467 ± 0.128 | 0.986 ± 0.010 |
> | MCPL-one+CL+Mask    | 0.271 ± 0.078 | 0.821 ± 0.050 | 0.534 ± 0.112 | 0.991 ± 0.006 |
> | BAS                 | 0.276 ± 0.078 | 0.823 ± 0.050 | 0.552 ± 0.097 | 0.995 ± 0.030 |
>
> **Object-level fidelity learned concepts referencing to "ground truth" (medical images)**
>
> | exp_names           | bert          | clip          | dinov1        | dinov2        |
> |---------------------|---------------|---------------|---------------|---------------|
> | MCPL-all            | 0.240 ± 0.065 | 0.787 ± 0.046 | 0.349 ± 0.069 | 0.977 ± 0.011 |
> | MCPL-one            | 0.250 ± 0.069 | 0.776 ± 0.044 | 0.329 ± 0.062 | 0.982 ± 0.010 |
> | MCPL-all+CL         | 0.251 ± 0.083 | 0.785 ± 0.040 | 0.339 ± 0.062 | 0.982 ± 0.011 |
> | MCPL-all+CL+Mask    | 0.264 ± 0.065 | 0.785 ± 0.045 | 0.353 ± 0.068 | 0.982 ± 0.010 |
> | MCPL-one+CL         | 0.247 ± 0.070 | 0.791 ± 0.047 | 0.376 ± 0.095 | 0.981 ± 0.012 |
> | MCPL-one+CL+Mask    | 0.262 ± 0.072 | 0.792 ± 0.037 | 0.407 ± 0.081 | 0.983 ± 0.009 |
> | BAS                 | 0.267 ± 0.079 | 0.823 ± 0.043 | 0.560 ± 0.080 | 0.987 ± 0.021 |
>
> * **The new t-SNE results in** Figure 7 demonstrate that the learned embeddings from both the mask-based ’ground truth’ and BAS show less disentanglement compared to ours, attributable to their lack of a specific disentanglement objective, such as the PromptCL loss in MCPL.
>
> ---
>
>  > _W.1 - The paper is not very clear to read. The idea seems to be straightforward, but the description of the method is a bit ambiguous…_
>
> * We thank the reviewer for pointing out the clarity issue in our manuscript. In response, we have **simplified certain descriptions in the methods section** to enhance clarity.
> * We also acknowledge our methods section combined method descriptions with preliminary motivational experiments, and this design might have disrupted the flow of the method description. To address this, we have **reorganised the methods section, including adjustments to Figure 4**, to ensure a more coherent and seamless presentation of our methods.

---

### Author Response · Authors · 2023-11-23
**Summary**

We thank the reviewers for their constructive feedback and patience with our extensive new experiments. We summarise the paper's strengths, reviewers' criticisms, highlight key new results and bring to attention issues that are noteworthy in our view:

---

**Strengths & reviewers’ support**:

We are delighted that the reviewers recognise the following **key strengths**:

* Our work **pioneers the novel task of mask-free text-guided learning for multiple prompts from one scene**, by introducing a **Multi-Concept Prompt Learning (MCPL)** method.
* The **motivational study** highlights the limitations of vanilla MCPL. In response, we've introduced **three innovative regularisation techniques** to enhance the accuracy of object-level concept learning.
* **Extensive experiments and analyses** show that our method outperforms competitive baselines and achieves results similar to the top mask-based method, Break-A-Scene, **underscoring the effectiveness of our mask-free approach**.
* We have compiled a **new dataset** for a comprehensive evaluation, involving both natural and biomedical images, **setting up a new benchmark for the field**.
* Overall, our approach not only enhances current methodologies but also **paves the way for novel applications**, such as facilitating knowledge discovery through natural language-driven interactions between humans and machines.

We appreciate the reviewers' recognition of our **method's novelty**, being the **first to achieve multi-concept extractions from a single image without relying on object masks** **as input**.

 > * R1(BYgx): “The paper targets important research areas.”
 > * R2(RxXV): “The proposed method leverages novel regularisation losses without relying on any groundtruth object segmentation”
 > * R4(wHAP): “A novel task … addresses the challenge ... which has not been previously explored”, “These effective techniques contribute to learning object-level information under image-level supervision.”

The reviewers acknowledge the submission's **well-structured and easy-to-follow** format, along with **well-designed experiments** and **extensive quantitative evaluations**, was also noted by the reviewers.

 > * R2(RxXV): “Overall, the paper is clearly structured and easy to follow. The motivational study introduces the problem and the current limitation in a systematic way”, “The experiments are well-designed …”, “Comprehensive analysis is conducted”.
 > * R3(cRiH): “The description of method and experiment section is polished and easy to understand.”, “Extensive experimental results ...”
 > * R4(wHAP): “validated through extensive quantitative analysis and evaluation”

---

**Main issues raised by the reviewers**:

We appreciate the reviewers for their helpful feedback, including pointing out typos and errors. The **primary critiques** were **R1(BYgx)**'s **quantification** concern and **R2(RxXV)**'s suggestion for a **baseline**. Other queries involved **ablation studies** to fully assess our method. We **conducted comprehensive new experiments, adding seven pages to the revised manuscript** to address all these points.

* **Extensive quantitative evaluation has been added** to include [Break-A-Scene](https://arxiv.org/abs/2305.16311) by Avrahami, O., et al., (SIGGRAPH 2023), as per **R2(RxXV)**'s suggestion. This top mask-based learning method now serves as a benchmark in our study. We updated Figures 5, 7, 8, 16-23, added Figures 10-12, and elaborated in Sections A.1 and A.2. With **160 additional single-GPU runs**, our approach is now quantitatively compared against **three baselines** (Textural Inversion, MCPL, and BAS) and qualitatively against **five more, thoroughly addressing the field's scope**.

* **Comprehensive** **ablation studies are added:**
    - **MCPL-diverse versus the full version of MCPL-one** with qualitative results in Figures 13 and 14 in Section A.3, in addition to Figure 11-12 in Section A.2, following suggestions of **R3(cRiH)** and **R4(wHAP)**.
    - **Learning with more than two concepts.** This is covered by Figure 11-12 in Section A.2, responding to **R4(wHAP)** about and **R2(RxXV)**'s query on mismatch learning.
    - **Composing with different strings from learning.** This is also covered in Figure 11-12 in Section A.2 addressing **R4(wHAP)'s question**.
    - **Simple prompt without adjectives.** We added Figure 15 in Section A.4 featuring an ablation study of the **impact of adjectives**, addressing **R2(RxXV)**'s inquiry.

---

**Mismatch between reviews & scores**: Lastly, we'd like to point out a significant discrepancy between **R1(BYgx)**'s feedback and the other three reviewers' comments. Specifically, **R1(BYgx)**'s concerns about clarity and novelty, contrast with the positive support from others. While we've addressed **R1(BYgx)**'s quantification concerns with substantial new results, we would be grateful if the reviewers could revisit their evaluation. We are eager to discuss further and clarify as much as needed.

---

### Meta-Review · Area_Chair_Gs9p · 2023-12-08

**Metareview:**

This paper introduces an innovative approach in textural inversion, focusing on learning multiple object-level concepts simultaneously from a single sentence-image pair. The Multi-Concept Prompt Learning (MCPL) framework is evaluated extensively, and its strengths and weaknesses are highlighted by the reviewers:

Strengths:

+ Innovative Framework for Textural Inversion: The MCPL framework is a novel approach that addresses the challenge of integrating multiple object-level concepts within a single scene, an area that has not been extensively explored before.

+ Effective Regularization Techniques: The introduction of three regularization techniques (Attention Masking, Prompts Contrastive Loss, and Bind adjective) effectively enhances the accuracy of word-concept correlation and refines attention mask boundaries.

+ Comprehensive Experimental Design: The experiments are well-designed and encompass both real-world categories and biomedical images, providing a broad evaluation of the framework's applicability.


Weaknesses:

- Clarity and Ambiguity in Method Description: The description of the method is somewhat unclear and requires multiple readings to be fully understood.


- Insufficient Explanation of Implementation Details: More details on prompt initialization and the determination of the number of prompts for MCPL are needed for clarity.

- Limited Demonstration of Composing Ability: The demonstrations to prove the composing ability of multiple concepts are insufficient and sometimes ineffective when the interaction between concepts is altered. (Only qualitative results presented)


- Generalization Concerns with New Dataset: The new dataset constructed for evaluating the framework primarily contains examples with only two distinct concepts, raising questions about the method's generalization capabilities.

- Lack of Comprehensive Quantitative Results: There is a notable lack of quantitative results that would allow for a fair comparison with other methods and a clearer understanding of this work's contribution. (Addressed in rebuttal)

- Omission of Recent Related Work: Some recent works in similar tasks, such as Break-A-Scene, are not mentioned in the related work section, missing a potential benchmark for comparison. (Addressed in rebuttal)

Given these insights, while the paper demonstrates innovation and potential through its framework and regularization techniques, addressing the noted weaknesses, particularly in terms of clarity, comprehensive quantitative analysis, and demonstration of composing ability, could enhance its impact and applicability in the field.

**Justification For Why Not Higher Score:**

See weaknesses.

**Justification For Why Not Lower Score:**

See strengths.

---

### Decision · Program_Chairs · 2024-01-16

Reject